

# Hydrological modeling of the Peruvian-Ecuadorian Amazon basin using GPM-IMERG satellite-based precipitation dataset

Ricardo Zubieta[1,2], Augusto Getirana[3,4], Jhan Carlo Espinoza[1,2], Waldo Lavado-Casimiro[5,2], Luis Aragon[2]

[1] Subdirección de Ciencias de la Atmósfera e Hidrósfera (SCAH), Instituto Geofísico del Perú (IGP), Lima, Peru

[2] Programa de Doctorado en Recursos Hídricos, Universidad Nacional Agraria La Molina, Peru

[3] Hydrological Sciences Laboratory, NASA Goddard Space Flight Center, Greenbelt, MD, USA

[4] Earth System Science Interdisciplinary Center, College Park, MD, USA

[5] Servicio Nacional de Meteorología e Hidrología (SENAMHI), Lima, Peru

*Correspondence to*: R. Zubieta (ricardo.zubieta@igp.gob.pe).

## Abstract

In the last two decades, rainfall estimates provided by the Tropical Rainfall Measurement Mission (TRMM) have proven applicable in hydrological studies. The Global Precipitation Measurement (GPM) mission, which provides the new generation of rainfall estimates, is now considered a global successor to TRMM. The usefulness of GPM data in hydrological applications, however, has not yet been evaluated over the Andean and Amazonian regions. This study uses GPM data provided by the Integrated Multi-satellite Retrievals (IMERG) (product/final run) as input to a distributed hydrological model for the Amazon Basin of Peru and Ecuador for a 16-month period (from March 2014 to June 2015) when all datasets are available. TRMM products (TMPA V7, TMPA RT datasets) and a gridded precipitation dataset processed from observed rainfall are used for comparison. The results indicate that precipitation data derived from GPM-IMERG correspond more closely to TMPA V7 than TMPA RT datasets, but both GPM-IMERG and TMPA V7 precipitation data tend to overestimate, in comparison to observed rainfall (by 11.1% and 15.7 %, respectively). In general, GPM-IMERG, TMPA V7 and TMPA RT correlate with observed rainfall, with a similar number of rain events correctly detected (~20%). Statistical analysis of modeled streamflows indicates that GPM-IMERG is as useful as TMPA V7 or TMPA RT datasets in southern regions (Ucayali basin). GPM-IMERG, TMPA V7 and TMPA RT do not properly simulate streamflows in northern regions (Marañón and Napo basins), probably because of the lack of adequate rainfall estimates in northern Peru and the Ecuadorian Amazon.



Keywords: GPM; Precipitation datasets; Hydrological modeling; Amazon/Andes; TRMM

## 1. Introduction

5 Satellite-based precipitation data have been widely used for hydrometeorological applications, such as hydrological modeling, especially in data-sparse regions like the Amazon River basin [Collischonn et al, 2008; Getirana et al, 2011; Paiva et al., 2013, Zulkafli et al., 2014; Zubieta et al., 2015]. Rainfall is extremely variable in both space and time, particularly over regions characterized by topographic contrast, such as the western Amazon Basin [Espinoza et al., 2009; Lavado et al., 2012]. In this region, the Andes Mountains contribute to high spatio-temporal variability of rainfall [Laraque et al., 2007, Espinoza et al., 2015]. To improve approximation and reduce uncertainty, detailed monitoring is needed using a high-density rain gauge network. Only a low-density rain gauge network is available in the Amazon basin (AB), however, which limits understanding of hydrological processes and hydrological modeling over the region [Getirana et al., 2011; Paiva et al., 2013]. Satellite-based datasets offer an alternative for modeling hydrological events, because they are uniformly distributed in both space and time. Their usefulness in Andean-Amazon basins and their applicability as input to hydrological models have been evaluated recently by comparing modeled and observed datasets. Results indicate that these datasets could be used for operational applications [Zulkafli et al., 2014; Zubieta et al., 2015]. This could improve streamflow modeling and relationships between model parameters in unmonitored basins.

Hydrological modeling and forecasting are still poorly developed in the Andean and Amazonian regions. It is important to improve this tool, especially because of an intensification of extreme hydrological events in the Amazon basin [Gloor et al., 2013], such as intense droughts in 2005 and 2010 [Marengo et al., 2008; Marengo et al., 2011; Espinoza et al., 2011] and severe floods in 2009, 2012 and 2014 [Espinoza et al, 2012; 2013; 2014]. Moreover, a high percentage of total annual precipitation can fall in just a few days, causing soil erosion, landslides [Zubieta et al., 2016]

In the last two decades, advances in satellite technology have improved rainfall estimation in much of the world [Huffman et al, 2007]. The Tropical Rainfall Measuring Mission (TRMM) Multi-satellite Precipitation Analysis (TMPA) precipitation dataset [Huffman et al, 2007] has been important to research and to many hydrological applications on Amazon regions, and there is consensus among studies using TMPA in Amazon regions [Collischonn et al, 2008; Getirana et al, 2011; Paiva et al., 2013, Zulkafli et al., 2014; Zubieta et al., 2015]. The TRMM mission ended in April 8, 2015, however, after the spacecraft depleted its fuel reserves (https://pmm.nasa.gov/trmm/mission-end). Despite the actual demise of TRMM, this is not the substantive issue for some products, such as TMPA and TMPA–RT, which are expected to run in parallel with the new GPM (Global Precipitation Measurement) satellite until mid-2017 [Huffman et al., 2015]. GPM, which was launched in February 2014, provides a new generation of rainfall estimation every three hours [Schwaller and Morris, 2011]. Recent studies highlight that the GPM Integrated Multi-satellite Retrievals (IMERG) estimations can adequately substitute TMPA estimations both hydrologically and statistically, even with its limited data availability [Liu, 2016; Tang et al., 2016].



The aim of this paper is to evaluate the use of rainfall estimates from the GPM-IMERG in obtaining streamflows over the Amazon Basin of Peru and Ecuador (ABPE) during a 16-month period (from March 2014 to June 2015) when all datasets are available. It provides a comparative analysis of the GPM-IMERG datasets and TRMM (TMPA RT and TMPA V7 products). For comparison with satellite estimations, a precipitation dataset from observed rainfall was developed by spatial

interpolation using the Peruvian National Meteorology and Hydrology Service (SENAMHI) network. Each precipitation dataset was used as input for the MGB-IPH hydrological model [Collischonn et al., 2007], which was recently adapted to ABPE [Zubieta et al., 2015]. The ABPE extends from the tropical Andes to the Peruvian Amazon, with elevations ranging up to 6,300 meters above sea level, a drainage area of 878, 300 km$^2$ and a mean discharge of around ~35,500 m$^3$/s at the Tabatinga station [Lavado et al., 2012]. The ABPE is located in the northwest of the AB (Fig. 1a), and its area corresponds

to 14% of the AB. It consists mainly of basins such as the Ucayali basin (southern of the ABPE), Marañón basin (Western of the ABPE) and Napo basin (Northern of the ABPE) (Fig. 1b).

## 2.   Datasets used in hydrological modeling

GPM and TRMM are satellite missions of the space agencies of the United States (NASA) and Japan (JAXA). They provide precipitation data derived from their products, which are evaluated in this study:

a)   GPM (product final IMERG-V03D) data at several of processing have been provided since March 2014 (http://pmm.nasa.gov/GPM), using raw satellite measurements and making the best estimate of global precipitation maps using combinations of observations and other meteorological data (http://www.nasa.gov/gpm). IMERG was designed to improve the limited sampling available from single low earth orbit (leo) satellites by using as many leo-satellites as possible, using geosynchronous-Earth-orbit (geo) infrared (IR) estimates to fill in gaps [Huffman et al.,

2015].  The temporal resolution of IMERG-V03D is half-hourly, and it has a 0.15-degree by 0.15-degree spatial resolution.

b)   TMPA 3B42 version 7 is obtained from a set of data provided by different satellite-based sensors between 1998 and April 2015, in both real and near-real time (TMPA 3b42 data are available at ftp://disc2.nascom.nasa.gov/data/TRMM/Gridded/3B42RT). TMPA has been essential for creating spatio-temporal

average levels that are appropriate for user applications, with very good results in climate and hydrological studies in recent decades [Huffman et al., 2007]. The usefulness of TMPA for hydrological modeling in the Amazon basin has been evaluated, for example, in Paiva et al., 2013; Zulkafli et al., 2014; and Zubieta et al., 2015.

c)   TMPA RT (real time) precipitation data are related to TMPA V7, but do not include calibration measurements of rainy seasons, which are incorporated more than a month after the satellite data.

(ftp://disc2.nascom.nasa.gov/data/TRMM/Gridded/3B42RT). As with TMPA V7, the final, gridded, sub-daily temporal resolution of TMPA RT is usually every three hours, with a 0.25-degree by 0.25-degree spatial resolution.

d)   To evaluate satellite-based datasets, a precipitation product was obtained using daily data series (PLU) from SENAMHI rainfall stations.  We collected daily rainfall data for 202 rain stations during the selected period. Quality control based on the Regional Vector Method (RVM) was used to select stations having the lowest probability of errors in their data





series [Hiez 1977; Brunet-Moret 1979]. Finally, 181 RVM-approved rainfall data series [distributed over 700,000 km$^2$] were selected, with data between March 2014 and June 2015 (Fig. 1b). The area with the highest data availability covers around 81% of the ABPE (19% without availability is mainly located in the northern region), where the largest distribution of rainfall stations is in the Andean regions, rather than Amazonian regions, of the Ucayali and Huallaga basins (the Huallaga is a sub-basin of the Marañón basin). For comparison, both regions with and without availability of rainfall data were considered for hydrological modeling. Rainfall observations subsequently were spatially interpolated to a resolution of $0.1^\circ \times 0.1^\circ$ by ordinary kriging, and a spherical semivariogram model was used to generate a gridded daily rainfall dataset. To use each precipitation dataset as input to the hydrological model, sub-daily data (for example, TMPA datasets have temporal resolution of 3 hours) were rescaled to a daily time step.

To evaluate model results, streamflow series from the SO-HYBAM Observatory (www.ore-hybam.org) and SENAMHI stations for the selected period were used; these were KM105 (KM), Mejorada (ME), Chazuta (CHA), Borja (BO), Bellavista (BE), Lagarto (LA), Pucallpa (PU), Requena (RE), San Regis (SR), Tamshiyacu (TAM) and Tabatinga (TAB) (Fig. 1b, Table 1). To describe climate characteristics, meteorological data from NCEP-DOE Reanalysis at surface level (Kanamitsu et al., 2002) were collected, including relative humidity, wind speed, solar radiation, air temperature and atmospheric pressure. Basin topography is derived from the Shuttle Radar Topography Mission (SRTM, version 2). Digital thematic maps correspond to vegetation and soil maps of Peru (http://www.fao.org) and a vegetation type map of Ecuador (http://sociobosque.ambiente.gob.ec/). A soil map of Ecuador (SECS-Ecuador, http: //www.secsuelo.org) and soil and land-use maps of Colombia (IGAC-Colombia, http://geoportal.igac.gov.co) were also considered. GPM-IMERG, TMPA V7, TMPA RT and PLU datasets were selected for the period corresponding to observed streamflows.

## 3. Methodology

The MGB-IPH model [Collischonn et al., 2007] has been used to simulate the hydrological behavior of the ABPE. It consists of modules for calculating soil water budget, evapotranspiration, flow propagation within a cell, and flow routing through the drainage network. A HRU (hydrological response unit) [Kouwen et al., 1993] approach is used to perform soil water balance by mean spatial classification of all areas with a similar combination of soil and land cover, which are associated with model parameters. To create HRUs, the watershed is divided into regular elements (cells), which are interconnected by channels. A parameter set is calculated separately for each HRU of each pixel, considering only one layer of soil [Collischonn et al., 2007]. The Muskingum-Cunge method is used for routing streamflows through the river network from runoff generated for different HRUs in the cells. Streamflows are adjusted for accuracy according to the stream reach length and slope. A detailed description of the MGB-IPH model is provided in Collischonn et al. [2007].

The comparison of precipitation datasets was performed in two steps: first, an analysis of monthly averages and detected rain



events at different precipitation thresholds (0.1, 1, 5, 10 and 20 mm/day) was conducted over the ABPE. The analysis was performed by computing the frequency bias index (FBI), probability of detection (POD), false alarm ratio (FAR), and equitable threat score (ETS) (see Table 2). These are calculated from a 2 x 2 contingency matrix composed of four parameters (a, b, c, d), where a is the number of observed rain events correctly detected, b is the number of observed rain

events not detected, c is the number of rainfall events detected but not observed (false alarms), and d is the sum of cases in which neither observed nor detected rain events occurred. FBI allows analysis of overestimation or underestimation of rain events, PDO provides information about sensitivity to not-detected and detected events, FAR is a function of false alarms, while ETS indicates the fraction of observed and/or detected rain events that were correctly detected.

Two performance coefficients were then used to evaluate the streamflow simulations: the Nash Sutcliffe (NS) coefficient,

and the difference between volumes calculated and observed (ΔV), shown in equations 1 and 2:

$$NS = 1 - \frac{\sum_{t=1}^{nt}(Q_{obs}(t)+Q_{cal}(t))^2}{\sum_{t=1}^{nt}(Q_{obs}(t)-\overline{Q_{obs}})^2} \qquad [1]$$

$$\Delta V = \frac{\sum(Q_{obs}(t))-\sum(Q_{obs}(t))}{\sum(Q_{obs}(t))} \qquad [2]$$

With Q_obs observed and Q_cal modeled streamflows. The range of efficiency lies from −∞ to 1, an efficiency of 1 (E = 1) corresponds to a perfect fit of modeled streamflow and observed data. While, an efficiency of lower than zero indicates that the mean value of the time series (observed) would have been a better predictor than the model. A Taylor diagram was used to provide a graphic summary of how closely a pattern (or a set of simulated streamflows) matches observed streamflows. In this diagram, the similarity among three statistical patterns is quantified according to the amplitude of their coefficient of

variation (CV %), correlation coefficient and centered root-mean-square difference (RMSD %) [Taylor, 2001]. This can be used to analyze the relative ability of hydrological models to simulate the spatial pattern of mean streamflow.

## 4. Results

### 4.1 Comparison of GPM-IMERG and other rainfall datasets

Total annual rainfall over the ABPE during the selected period is shown in Figs. 1c-f, using all four precipitation products.

The satellite-based datasets (GPM-IMERG, TMPA V7 and TMPA RT) overestimate observations (PLU) during this period (by 11.1%, 15.7% and 27.7 %, respectively). As Figs. 1c-f show, the satellite-based products present similar spatial distributions. These products are comparable to the PLU over a) the Andean regions (for this paper, the Andean and Amazon regions are considered to be above and below 1500 meters above sea level, respectively, see Fig 1b), with precipitation mainly between 500 and 1500 mm/year, and b) the northern Amazon regions (3.0°S-6.0°S), with precipitation between 2000

and 3000 mm/year. There are some spatial differences over the southern Amazon regions. This can be attributed to greater




uncertainty of the PLU dataset, however, because there are fewer rainfall stations in those regions, particularly the eastern Ucayali basin (Fig 1b).

A comparison of monthly rainfall over the Ucayali and Huallaga river basins (at the Requena and Chazuta stations) with satellite-based precipitation data during the selected period is shown in Figs. 2a and 3a, where the spatial distribution of rainfall stations is greater in the Andes region than the Amazon region. The TMPA V7 and GPM-IMERG datasets are very similar to each other in the Ucayali and Huallaga river basins. A monthly rainfall analysis shows that TMPA V7 and GPM-IMERG tend to underestimate dry-season rainfall in the Ucayali basin (April to September), by 10.6%, compared to PLU dataset (Fig. 2a). Meanwhile, both datasets tend to slightly overestimates wet-season rainfall by 3% compared to PLU dataset. This overestimation is larger than obtained by TMPA V7 or GPM-IMER when TMPA RT is analyzed (17.5%). Moreover, GPM-IMERG, TMPA V7 and TMPA RT datasets tend to underestimate dry and wet-season rainfall in the Huallaga basin by 30.7%, 28.2% and 26.2%, respectively, compared to PLU (Fig. 3a).

Building on the average number of total days of rain events (456), the number of rain events correctly detected (~ 20%) is similar for each satellite precipitation dataset, compared to PLU dataset over the Ucayali and Huallaga basins (Figs. 2b and 3b). The average number of events correctly and not correctly detected is also consistent—that is, all precipitation datasets are clearly better at identifying low- and moderate-precipitation events (1 - 5 mm/day) than the number of high- and very low-precipitation events (higher than 5 mm/day and lower than 1 mm/day respectively) (Figs. 2b-c and 3b-c). Average FBI values obtained for all datasets indicate a low capacity in the detection of rain events greater than 5 mm/day, producing FBI values varying mainly between 1 and 2 in the Ucayali and Huallaga basins. This differs substantially from optimal conditions (~1) (Figs. 2f and 3f). This variation is due to the high number of rain events that were not correctly detected (~80%) (Figs. 2c and 3c).

Average POD values for all datasets indicate a moderate probability of detection (POD less than 0.55) rain events greater than 5 mm/day, this probability decreases to ~0.2 for others events in the Ucayali and Huallaga basins (Figs. 2g and 3g). The average number of events correctly and not correctly detected is also consistent—that is, all precipitation datasets are clearly better at identifying precipitation events between 1 and 5 mm/day. The low probability of detection is consistent with the fraction of rain events which were correctly detected (ETS) (Figs. 2i and 3i). This is due to a high rate of false alarms (FAR) between ~0.7 and ~0.9 for rain events higher than 5 mm/day and lower than 1 mm/day for all satellite precipitation datasets in both the Ucayali and Huallaga basins (Figs. 2h and 3h).

The limited ability to represent rainfall events of more than 5 mm/day using satellite-precipitation datasets (GPM-IMERG, TMPA V7, TMPA RT), compared with PLU dataset (Figs. 2g and 3g), may be due to slight overestimation (in the Ucayali basin) or high overestimation (in the Huallaga basin) identified mainly during the wet season (Figs. 2a and 3a). Events exceeding 5mm/day are more likely to occur during that period.



## 4.2 Streamflow simulation

In order to optimize the simulation of streamflows from precipitation datasets, different parameter sets were assigned to each basin in ABPE during calibration. The analysis by sub-basin is more reliable than assigning the same parameter set to the entire basin [Zubieta et al., 2015]. Simulated streamflows were compared to observations at 11 gauging stations (Fig. 1b,

Table 3). The Ucayali and Huallaga basins (with greater availability of rainfall gauges) and the northern region of ABPE (without rainfall gauge availability) were considered in the comparative analysis. In general, streamflows obtained from all satellite-based precipitation datasets show the same spatial pattern as those obtained by using PLU (Figs. 4a-b) and are similar to those obtained by Zubieta et al. [2015]. This study shows that GPM-IMERG can also be a helpful alternative source of data (similar to TMPA V7 and TMPA RT) for rainfall–runoff simulation in areas where there is a lack of

conventional rainfall data, such as the Andean-Amazon regions of the Ucayali basin. The performance analysis over the equatorial regions does not agree well with observed streamflows (NS lower than 0.60), probably because of the lack of adequate rainfall estimates. Similar results are also obtained using the satellite precipitation datasets TMPA V7 (Fig. 4c) and TMPA RT (Fig. 4d) in the hydrological modeling.

Figs. 5a-f shows the ability of the MGB-IPH model to simulate observed streamflows using TMPA V7, TMPA RT, GPM-

IMERG and PLU precipitation datasets. Simulated streamflows match observations at six stations a) Chazuta (CHA) b) Km105 (KM), c) Lagarto (LA), d) Mejorada (ME), e) Pucallpa (PU), and e) Requena (RE). The location of each dataset on the plot quantifies how closely modeled streamflows match observed streamflows in terms of CV, correlation coefficient and RMSD.

Fig. 5a shows a Taylor diagram for the Chazuta station (Huallaga basin), where modeled streamflows from PLU dataset

agree better with observed streamflows (r=0.84, p<0.01), RMSD error (30%) and CV of 29%) than do those using data from satellite products (TMPA RT, TMPA V7 and GPM-IMERG). Analysis of the two smallest sub-basins (in the Ucayali basin) controlled at the KM (Fig. 5b) and ME (Fig. 5c) stations shows a correlation pattern of r= ~ 0.9 with RMSD of ~40% at KM and 24-40% at ME (Fig. 5b-c). These results indicate that the streamflows from PLU and TMPA RT are more similar to observed streamflow series mainly at ME, with RMSD lower than 30%. Both streamflow series at KM and ME have a

high CV (40%-80%), due to rainfall seasonality.

Analysis of the largest sub-basins (in the Ucayali basin) controlled at the LA, PU and RE stations shows greater similarity among them for the four streamflow series obtained from precipitation datasets (Fig. 5d-f). Their significant correlation patterns are between 0.8 and 0.9 (r > 0.9 at the PU station), and RMSD are mainly between 20% and 25% (PU and RE). It should be noted that streamflow data series have a lower CV in the larger sub-basins, such as LA, with CV of 55% (drainage

area of 191,400 km$^2$); PU, with CV ~ of 42% (drainage area of 260,400 km$^2$); and RE, with CV of ~40% (drainage area of 350,200 km$^2$). This could be due to weaker seasonality of rainfall in the northern part of the basin. As for simulations using satellite-based precipitation datasets, the correlation between simulated and observed streamflows is mainly between 0.6 and 0.9, and RMSD is relatively high (20% - 40%), suggesting that a hydrological model using these datasets can represent seasonal streamflows.



PLU dataset used as input to the hydrological model produced good results at the KM 105 (NS = 0.82 and ΔV =0.33%) (Fig. 6a), Mejorada (NS = 0.89 and ΔV = 4.2%) and Lagarto (NS = 0.74 and ΔV =-9.52%) stations in the Ucayali basin. This indicates its ability to represent extreme values (peak flow) with a low percentage of relative volume error (ΔV < 10 %). However, the model's performance is low at the Pucallpa and Requena stations (NS < 0.51 and ΔV ~ 10%), where its

predictions are not accurate. The low performance (NS < 0.60) is associated with drainage areas greater than the approximate threshold value of 200,000 km$^2$ in the Ucayali basin. This could be due to greater uncertainty in the spatial distribution of rainfall in the Ucayali and Huallaga basins (northern region of the ABPE), because there are fewer rainfall stations in these regions. The Peruvian Andes are currently more instrumented than the Amazon regions (see Fig. 1b).

To analyze the ability of GPM-IMERG datasets in hydrological modeling were analyzed hydrographs for the Ucayali basin monitored at Km 105 station (Fig. 6b), for comparison, streamflows from PLU, TMPA V7 and TMPA RT datasets were also considered (Fig. 6c-d). Visual analysis of the hydrographs shows that simulated streamflows using GPM-IMERG for the selected period agree fairly well with observed streamflows for the KM 105 station. Although the Nash–Sutcliffe efficiency coefficient is generally acceptable (NS = 0.90 and ΔV = -0.25%), as shown in Fig. 6b, there is a slight streamflow

overestimation during the wet season, which could be due to overestimation of rainfall during this wet season. Other results indicate that the model's performance is minimally acceptable in comparison to observed streamflow at the Pucallpa (NS = 0.61, ΔV = -17.2%) (Fig. 6g), and Mejorada stations (NS = 0.61, ΔV = -18.5%). For the other stations, performance within the basin is less than zero.

Similar results were observed using TMPA V7 and TMPA RT, which reproduce the seasonal streamflow regime with similar performance at the KM 105 (NS =0.80 and ΔV = -2.78%, NS =0.68 and ΔV = 11.5%, respectively) (Figs. 6c-d) and Pucallpa (NS =0.60 and ΔV = -17.8%, NS =0.89 and ΔV = -8.3%, respectively) stations in the Ucayali basin (Figs. 6h-i).

### 5. Concluding Remarks

Three satellite-based precipitation datasets (GPM-IMERG, TMPA V7, and TMPA RT) were evaluated against a rain-gauge-based dataset (PLU) obtained by spatial interpolation over the Amazon basin of Peru and Ecuador. Each dataset was used as inputs for the MGB-IPH hydrological model to simulate streamflows for a 16-month period (from March 2014 to June 2015) in the Ucayali, Huallaga, Marañón, Napo, Amazonas and Solimões river basins.

GPM-IMERG and TMPA V7 show high temporal and spatial similarity with PLU in the Ucayali basin, but they tend to
underestimate PLU in the Huallaga basin during the wet season of the 2014-2015 period. TMPA RT tends to overestimate for the Ucayali basin, compared to other precipitation datasets (PLU, TMPA V7, GPM-IMERG), while it is more similar to other satellite-based precipitation datasets (TMPA V7, GPM-IMERG) in the Huallaga basin.

GPM-IMERG datasets shows greater similarity with TMPA V7 than TMPA RT. This indicates that GPM-IMERG estimates are more similar to TMPA V7 both spatially and temporally when used as input for hydrological modeling over Andean and



Amazon basins. On average, rain event detection coefficients also suggest that GPM-IMERG, TMPA V7 and TMPA RT are similar to PLU in the number of rain events correctly detected (~20%) for the Ucayali and Huallaga basins.

In general, the performance of the model when using GPM-IMERG datasets indicates that these data are useful for estimating observed streamflows in Andean-Amazonian regions (Ucayali basin, southern regions of the Peruvian and Ecuadorian Amazon Basin). These results are similar to those obtained from TMPA V7 estimates by Zubieta et al. [2015] for the 2003-2009 period. Streamflows obtained from GPM-IMERG, TMPA V7, TMPA RT datasets show the same spatial pattern as those obtained by using PLU, (low and high performance in the northern and southern regions of the ABPE, respectively). The ability to represent seasonal streamflows for the southern region using these four precipitation datasets is corroborated by statistical evaluation. Low performance of the model in the southern region is mainly related to the lack of adequate rainfall estimates, because it is consistent with estimated streamflows.

**Acknowledgements**

The authors would like to thank SENAMHI for providing observed precipitation data series and the SO-HYBAM observatory for providing discharge data [www.ore-hybam.org]. The authors also acknowledge GSFC/DAAC and PMM [NASA] for providing TMPA [http://disc.sci.gsfc.nasa.gov/ /data/TRMM/Gridded/] and GPM-IMERG [https://pmm.nasa.gov/data-access/downloads/gpm] datasets respectively. Finally, RZ and JCE wish to acknowledge the PNICP-Peru for providing funds through Contract 397-PNICP-PIAP-2014.

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





**Table 1.** Characteristics of streamflow gauging stations in the Amazon basin of Peru and Ecuador: Altitude, river, drainage area, annual mean streamflow (Q mean), maximum streamflow (Q max) and minimum streamflow (Q min) in m3/s.

| N | Station | Altitude | River | Area (Km 2) | Q medio (m3/s) | Q max (m3/s) | Q min (m3/s) |
|---|---------|----------|-------|-------------|----------------|--------------|--------------|
| 1 | Km 105 (KM) | 2275 | Ucayali | 9635 | 98 | 446 | 30 |
| 2 | Mejorada (ME) | 2799 | Ucayali | 16930 | 186 | 651 | 76 |
| 3 | Chazuta (CHA) | 226 | Marañon | 68685 | 3430 | 8921 | 936 |
| 4 | Borja (BOR) | 163 | Marañon | 92302 | 6123 | 13250 | 1931 |
| 5 | Bellavista (BE) | 90 | Napo | 100169 | 9338 | 13110 | 4654 |
| 6 | Lagarto (LA) | 200 | Ucayali | 191428 | 6194 | 30460 | 1292 |
| 7 | Pucallpa (PU) | 141 | Ucayali | 260418 | 10833 | 21830 | 3714 |
| 8 | Requena (RE) | 94 | Ucayali | 350215 | 13669 | 20910 | 4088 |
| 9 | San Regis (SR) | 92 | Marañon | 359883 | 20119 | 26610 | 9071 |
| 10 | Tamshiyacu (TAM) | 88 | Amazon | 682970 | 37380 | 53840 | 15000 |
| 11 | Tabatinga (TAB) | 62 | Solimões | 878141 | 45384 | 62190 | 19700 |





**Table 2.** Summary of rain event detection coefficients.

| Coefficient | Name | Equation* | Range | Optimal score |
|---|---|---|---|---|
| FBI | Frequency bias index | FBI = (a+b)/(a+c) | 0 - ∞ | 1 |
| POD | Probability of detection | POD = a/(a+c) | 0 - 1 | 1 |
| FAR | False alarm ratio | FAR = c / (a+c) | 0 - 1 | 0 |
| ETS | Equitable threat score | ETS = (a+He)/(a+b+c-He) | - ∞ to 1 | 1 |

* He = (a+b). (a+c)/N where N is the total number of estimates



**Table 3.** Summary of modeling results at 11 gauging stations in the Amazon basin of Peru and Ecuador (to Tabatinga station in Brazil).

| N | River | Station | OBSERVED RAINFALL (PLU) | | GPM-IMERG | | TMPA V7 | | TMPA RT | |
|---|-------|---------|------|------|------|------|------|------|------|------|
| | | | NS | Δ V | NS | Δ V | NS | Δ V | NS | Δ V |
| 1 | Ucayali | Km 105 (KM) | 0.82 | 0.33 | 0.90 | -0.25 | 0.80 | -2.78 | 0.68 | 11.55 |
| 2 | Ucayali | Mejorada (ME) | 0.89 | 4.2 | 0.61 | -18.5 | 0.61 | -17.01 | 0.75 | -6.49 |
| 3 | Ucayali | Chazuta (CHA) | 0.37 | -18.27 | -0.26 | -31.96 | -0.37 | -33.51 | -0.02 | -29.55 |
| 4 | Ucayali | Borja (BOR) | ----- | ----- | -3.94 | -47.98 | -3.09 | -42.39 | -3.91 | -47.53 |
| 5 | Ucayali | Bellavista (BE) | ----- | ----- | -2.17 | -7.14 | -18.24 | -32.64 | -20.93 | -35.46 |
| 6 | Marañon | Lagarto(LA) | 0.74 | -9.52 | 0.71 | -0.13 | 0.80 | -0.49 | 0.81 | -0.18 |
| 7 | Marañon | Pucallpa (PU) | 0.48 | -8.1 | 0.61 | -17.2 | 0.60 | -17.80 | 0.89 | -8.3 |
| 8 | Marañon | Requena (RE) | 0.51 | -10.6 | -3.75 | -23.59 | -7.71 | -33.28 | -5.33 | -23.32 |
| 9 | Napo | San Regis (SR) | ----- | ----- | -5.40 | -24.82 | -5.68 | -25.59 | -4.90 | -24.72 |
| 10 | Amazon | Tamshiyacu (TAM) | ----- | ----- | -24.51 | -32.22 | -33.32 | -37.57 | -28.19 | -33.19 |
| 11 | Solimões | Tabatinga (TAB) | ----- | ----- | -3.85 | -10.28 | -12.88 | -19.51 | -5.21 | -10.74 |




**Figure 1.** **(a)** Location of the Amazon basin in South America, **(b)** the Western Amazon basin, gauging and rainfall stations used in this work, intermittent line represents main isohypse 1500 m.a.s.l. Total annual precipitation estimated from **(c)** observed rainfall-PLU, **(d)** GPM-IMERG, **(e)** TMPA V7, **(f)** TMPA RT over the Amazon basin of Peru and Ecuador.





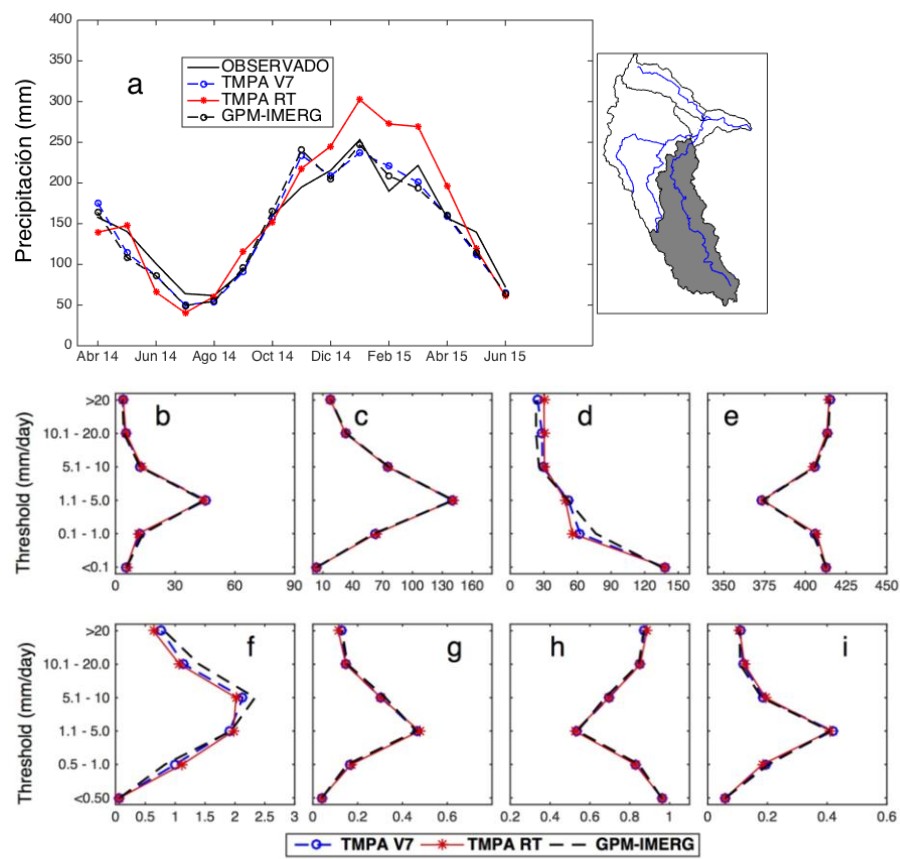

**Figure 2. (a)** Basin-average monthly rainfall for each precipitation dataset in the Ucayali basin up to Requena station, **(b)** the number of observed rain events correctly detected, **(c)** the number of observed rain events not correctly detected, **(d)** the number of rain events detected but not observed (false alarms), **(e)** the sum of cases when neither observed nor detected rain events occurred, **(f)** coefficient frequency bias index – FBI, **(g)** probability of detection-POD, **(h)** false alarm ratio – FAR, and **(i)** equitable threat score-ETS.





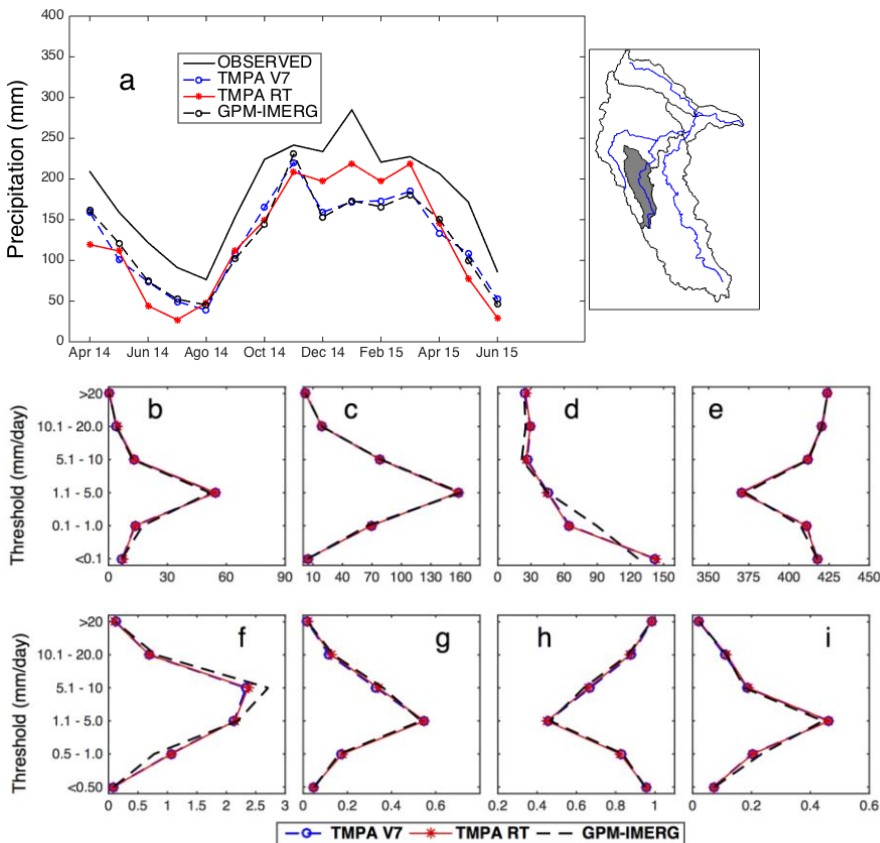

**Figure 3.** **(a)** Average monthly rainfall for each precipitation dataset in the Huallaga basin up to the Chazuta station, **(b)** the number of observed rain events correctly detected, **(c)** the number of observed rain events not correctly detected, **(d)** the number of rain events detected but not observed (false alarms), **(e)** the sum of cases when neither observed nor detected rain events occurred, **(f)** coefficient frequency bias index – FBI, **(g)** probability of detection-POD, **(h)** false alarm ratio – FAR, and **(i)** equitable threat score-ETS.





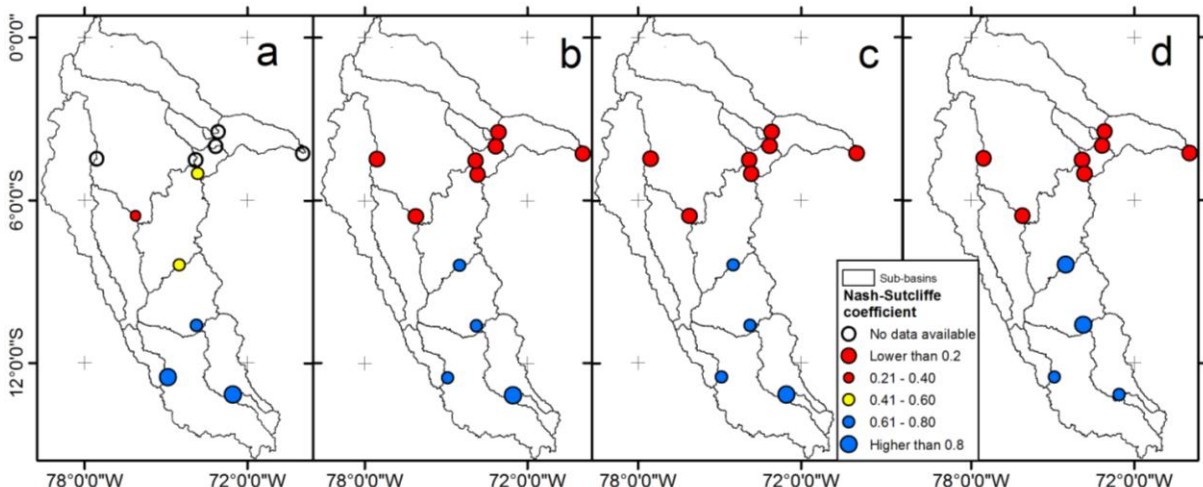

**Figure 4.** Nash–Sutcliffe efficiency coefficients map for simulations using: **(a)** Observed Rainfall (PLU), **(b)** GPM-IMERG, **(c)** TMPA V7 and **(d)** TMPA RT rainfall data.




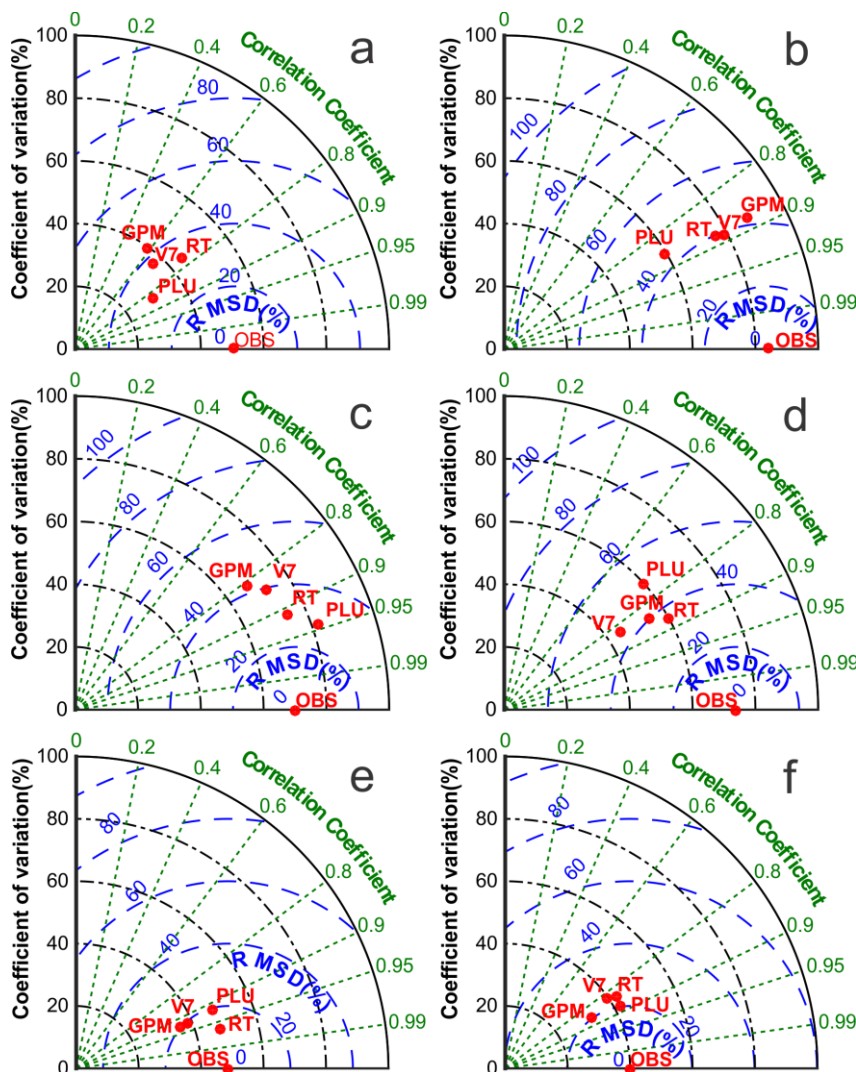

**Figure 5.** Taylor diagrams displaying a statistical comparison (coefficient of variation (%), the root mean square difference (%) and correlation coefficient) between observed streamflows and modeled streamflows from four precipitation datasets (TMPA V7 (V7), TMPA RT (RT), GPM-IMERG (GPM), observed rainfall (PLU)) for six basins controlled at stations: **a)** Chazuta (CHA), **b)** Km105 (KM), **c)** Mejorada (ME), **d)** Lagarto (LA), **e)** Pucallpa (PU), and **f)** Requena (RE).



**Figure 6.** Observed and simulated streamflow hydrographs at KM 105 station from March 12, 2014, to June 30, 2015, using precipitation datasets: **(a)** Observed rainfall, **(b)** GPM-IMERG, **(c)** TMPA V7, and **(d)** TMPA RT, **(e)** Location of the drainage area controlled at the KM station. Observed and simulated streamflow hydrographs at the Pucallpa station from March 12, 2014, to June 30, 2015, using precipitation datasets: **(f)** Observed rainfall, (g) GPM-IMERG, **(h)** TMPA V7, **(i)** TMPA RT; **(j)** Location of the drainage area controlled at the Pucallpa station.