# Peer review of "Hydrological modeling of the Peruvian-Ecuadorian Amazon basin using GPM-IMERG satellite-based precipitation dataset"

_Hydrology and Earth System Sciences, 2016_

## Referee Comment (RC1) · Anonymous Referee #1 · 12 Jan 2017

This paper examines the potential of the GPM-IMERG satellite-based precipitation dataset for obtaining reasonable model discharge estimates over the Peruvian-Ecuadorian Amazon basin. The MGB-IPH hydrological model, which has been used extensively to study the hydrological functioning of the Amazon basin, has been extended Westward and this study focuses on this region. In addition, several standard (commonly used) datasets, TMPA, are used and the results are compared with those using the GPM-IMERG data. These results are also compared to a purely rain-gauge based driven set of simulations. The results are fairly convincing in that the model is able to give quite reasonable discharge simulations over the period considered (for which the coverage of the different datasets overlap in time). The results are solid and

well presented: I also appreciate the concise nature of the paper for the most part: several details are glossed over and require a bit more discussion in the paper. I have highlighted some of these in my comments.

P.3 L.22: item b: Does this dataset (TMPA-V7) include information from the rain gauges shown in Fig.1 (all of them, half of them, just a few of them, none of them?). The authors note that TMPA-v7 and GPM-IMERG are the most similar...can the authors discuss a bit why this is so? I assume they use much of the same satellite and rain gauge data...?

P.3 L.5: Two questions related to details: could the observed rainfall have been interpolated to a 0.25x0.25 grid for better comparison with the TMPA data sets? 0.15x0.15 for the GPM-IMERG dataset? And in regions of mountainous terrain, oftentimes consideration of the altitude-rainfall gradient relationship is critical for spatially distributing rainfall onto a grid. Was this information included in the interpolation? Do the authors think this effect is important in this region?

P.4 L.9: MGB uses the rainfall aggregated to a daily time step. So, it seems that one of the main possible advantages of the GPM-IMERG (30 minute time step) datatset compared to the TMPA-V7 (3 hourly) for computing runoff generation by MGB is lost, especially since the convective nature of the rainfall is likely better resolved, in theory at least, using a 30 minute time step. Is there anything in MGB that can take advantage of the diurnal temporal distribution of the rainfall? If not, I think the authors should at least comment that hydrological models using sub-diurnal time steps might have larger differences between the TMPA and GPM-IMERG datasets owing to their different temporal resolutions...Can the authors comment on this?

P.5 L.1: How were these thresholds selected? Are they based on some sort of statistical analysis?

P.5 L.25: The text "overestimate observations"...should likely be modified to something like "produce overestimates compared to observations".

P.6 L.10-11: The authors have reported that the satellite-based datasets underestimate the dry and wet season rainfall much more for the Huallaga basin compared to the Ucayali basin: do the authors have any insights as to why this is?

P.6 L.18-20: While I am not surprised that detection of light events was difficult, why do the products have such difficulty predicting strong rainfall events? I might have (perhaps erroneously or naively I admit) assumed that such events might be better detected. Can the authors comment on this? For example, are the strongest events occurring in high altitude/mountainous regions which are more difficult to detect? Is the smoothing to daily averages related to this problem?

P.7 L.3: The calibration is glossed over a bit it seems: what set of parameters were calibrated? Were parameters calibrated separately for each precipitation dataset?. Also, no information is given on the quality or calibration of the MGB evaporation. Is it significant compared to the rainfall? Are there non-negligible compensating errors (evaporation bias might offset rainfall or discharge errors/biases)? A short discussion is needed.

P.9 L.8-9: The authors state that seasonal streamflows in the southern region are well modeled using the satellite datasets, and indeed the results support this conclusion. But in the northern part of the Western Amazon basin/region, the results seem to indicate that satellite products are not useful for obtaining streamflows from hydrological modeling: so this implies that further progress is still required.. It think this should also be stated in the conclusions.

P.9 L2.: A 20% detection rate seems low. Can the authors put this into some sort of context for the reader (e.g. is 20% indeed a reasonable value in this region, or would one hope to have 50%? or a higher value?). Also, what are the implications of these statistics and their impact on the MGB model simulations? Can a lower, say 10% detection rate, be assumed to be able to produce reasonable Nash scores? Or perhaps there is no clear relationship between these scores and the modeled discharge quality?

This is not clear. Can the authors comment on this?

P10 L.15-19: In Fig.4d, it is seen that the poorest Nash scores are in the northern part of the region shown: looking at Fig.4a (or Fig.1b), we see that there are relatively few observations in this region. But, this is where one would hope to benefit the most from a satellite product, but it seems this is not the case. In L.15-19 it is stated that such products hold promise for operational applications in data sparse regions, but it doesn't seem to be the case here? Can the authors comment a bit more or perhaps modify this slightly (seemingly) over-optimistic text?

General: the English is fairy good. My only comment is that occasionally words end in s which shouldn't or vice versa. Also, I saw quite a few instances where an article or "the" was missing. But aside from that, the text is in good shape. I would recommend tidying that up before publication: it should not be too much work for the authors.

---

## Referee Comment (RC2) · Anonymous Referee #2 · 13 Jan 2017

This relatively short paper presents a preliminary assessment of the i-merg rainfall product in the Peruvian and Ecuadorian parts of the Amazonian Basin. I-merg is the new high resolution gridded product associated with the GPM mission , launched in 2014. This paper provides an illustration of the potential of this new product for hydrological applications in a region which high altitudes gradients (tropical andes) where the rain gauge network is very sparse

General :

The paper is clearly written . The study is not very original and the scientific impact of the results is limited because of the scarcity of the available 'reference' rain gauge network and the short study period. However the paper provides interesting information

on the behavior of i-merg over the Amazonia-Andes region and its potential use as a forcing field for hydrological models.

I believe this study would be worthy of publication ; however some additional details on the methodology (see below) and a more in depth discussion on the limitations of the data, method and results is needed .

DETAILED COMMENTS

1- Scarcity of the rain gauge network and impact on the comparison results

The authors had to work with a very scarce and unevenly spread network – (scarcity which makes satellite information all the more attractive) . They acknowledge briefly that the small number of gauges in part of the basin might explain the discrepancies between the sat/ground rainfall products and between the simulated/observed discharge , but there is no attempt to quantify the uncertainty in the ground rainfall product.

- the authors should elaborate on the abilty of their ground rainfall product PLU to reproduce the rainfall gradients in the mountaneous part of the basins. Is altitude taken into account in their interpolation method and if yes how and was the quantitative uncertainty assessed ?

- The authors have used krigging to provide a product at the $0.1°$ resolution over the 700000 km2 basin, from a total of 181 gauges.

o It would be informative to know what the de-correlation distance of the variogram model is .

o Is anisotropy considered when interpolating in the montaneous areas ?

o Also as kriging provides the estimation variance, the authors could provide a map showing the expected quality of the ground product (for instance ratio of kriging std over rainfall estimate for one day or an average over the season)

- The authors provide comparison of satellite/ground product for basin average ; they

should indicate what are the results when comparing only over the grid points that contain a gauge (or are within a short distance from gauges).

- Given the points above, how certain are you that satellite products are overestimating rainfall (p 5 -section 4.1) rather than the ground based product underestimating it.

2 – information on model calibration and sources of uncertainty in the model run.

Section 4.2 :

- A description of the model configuration is lacking – the size of the HRU – and a discussion on whether or not it allows to take advantage of the products improved spatial resolution is missing.

- How was the model calibrated and on which period/data sets ?  is the model re-calibrated for each rainfall forcing ? if not/yes, why ?

- One of the benefit expected from new rainfall product like i-merg is their improved space/time resolution compared to coarser products. This important point is not discussed in the study. As the model is run a daily time step the beneft of improved time resolution cannot be assessed, however the authors could investigate the impact of the $0.1°$ grid provided by i-merg. For instance by smoothing or under sampling the product to a coarser resolution ($0°5$ for instance). And comparing the simulated discharge for both $0°1$ and a coarser spatial resolution.

- Rainfall is not the only source of uncertainty in the simulated discharge ; the ability of the chosen model to represent the hydrological processes in the studied region, especially in the mountainous sub-basins, should be discussed. Other sources of uncertainty –among them model parameters estimation- that might impact the results should also be mentioned and if they have been quantified, the information should be provided.

3 – information on satellite products version

- importantly I could not find the information on the version of i-merg which has been used here.

- Since there were several releases of i-merg since the launch and given that i-merg (just like 3B42) is provided with a gauge calibrated version and an un-calibrated or RT version, both should be tested here. For a fair comparison with 3B42 RT and v7.

- P3 – line 15 to 35 – The degree of information on products should be the same for both i-merg and 3B42 : number /type of contribution satellites, basic description of the stimation method.

- 3B42 (and i-merg calibrated version) use some gauges from weather services – Could you check which are the gauges used here were included in TMPA/i-merg correction stage ?

Other minor corrections :

-In the introduction and throughout the text there seem to be some confusion between i) the satellite themselves (TRMM or GPMcore satellite) , ii) satellite constellations (the GPM international program includes the NASA-JAXA GPM core satellite -and TRMM while it was still going- and a constellation of other satellites from various agencies) and iii) the rainfall products which are derived from this satellite constellations and do not depend on one particular satellite (TMPA 3B42 can be run without the TRMM satellite itself. . . i-merg could be run without GPMcore if necessary ) .

Exemple : intro p2 line 12-13 : 'satellite data sets . . ...are uniformly distributed in space and time' - Product like 3B42 or i-merg are provided on a regular space-time grid however the microwave satellite information itself is provided with a very irregular sampling and depends on individual orbits. . .. And as for gauges interpolation is done to prvide a final regularly gridded product.

P 2 – line 30-32 – confusion between GPMcore single satellite launched in 2014, and the GPM multi satellite/multiagencies constellation . . ...

Also : The improved resolution capacity of the latest products does not come from a specific satellite (though some members of the GPM constellation such as TRMM and more recently Megha-Tropiques , provide additional sampling specifically in the Tropics) but from the overall sampling capacity of the whole constellation. This should be mentioned more clearly in the intro.

Looking forward to see an improved version of this very interesting work.

---

## Author Comment (AC1) · 15 Feb 2017

**We would like to thank the referee for his useful comments. Please find bellow our answers.**

1.   P.3 L.22: item b: Does this dataset (TMPA-V7) include information from the rain gauges shown in Fig.1 (all of them, half of them, just a few of them, none of them?). The authors note that TMPA-v7 and GPM-IMERG are the most similar...can the authors discuss a bit why this is so? I assume they use much of the same satellite and rain gauge data...?

GPM is an international US/Japanese Earth science mission involving NASA and JAXA, respectively. The GPM mission improved and expanded on TRMM. GPM and TRMM provide precipitation data derived from different passive microwave (PMW) sources used in IMERG and TMPA, respectively [Huffman et al. 2015], including: Sounder for Atmospheric Profiling of Humidity in the Intertropics by Radiometry (SAPHIR), Advanced Technology Microwave Sounder (ATMS), Atmospheric Infrared Sounder (AIRS), Cross-Track Infrared Sounder (CRIS), and TRMM Combined Instrument (TCI) algorithms (2B31). They also include TRMM Microwave Image (TMI, data ended on 8 Apr 2015), GPM Microwave Imager (GMI), Advanced Microwave Scanning Radiometer for Earth Observing Systems (AMSR-E), Special Sensor Microwave Imager/Sounder (SSMIS), Microwave Humidity Sounder (MHS), Special Sensor Microwave Imager (SSM/I), Advanced Microwave Sounding Unit (AMSU), Operational Vertical Sounder (TOVS) and microwave-adjusted merged geo-infrared (IR).

TMPA 3B42 version 7 is obtained from the preprocessing of data provided by different satellite-based sensors between 1998 and April 2015, in both real and near-real time (TMPA 3b42 data are available at ftp://disc2.nascom.nasa.gov/data/TRMM/Gridded/3B42RT). The 3B42 algorithm (every three hours) combines precipitation estimates from TMI, AMSR, SSMIS, SSM/I, AMSU, MHS, TCI, *MetOp-B* and IR. After the preprocessing is complete, the 3-hourly multi-satellite estimations are summed for the month and combined with monthly rainfall obtained from Global Precipitation Climatology Centre (GPCC), which uses ground-based precipitation. The last step is to scale each 3-hourly rainfall estimate for the month to sum to the monthly value (for each pixel separately, 0.25-degree by 0.25-degree spatial resolution).

Both TMPA and GPM-IMERG adopt the Global Precipitation Climatology Centre (GPCC) monthly rain gauge analysis (Huffman et al. 2014).   The Monitoring Product is represented on internationally exchanged meteorological data i.e. gauge observations from world-wide 6,000 to 7,000 stations (see next figure, *(*Schneider et al., 2014). The average gauge density is about 2 gauges per 2.5° by 2.5° lat/long grid box only. Building upon the figure of rainfall stations and lat/long grid *(*Schneider et al., 2015), it very probably that 105 rainfall stations used in our study were considered by GPCC calculations.

[Figure]

*Spatial distribution of monthly in-situ stations with a climatological precipitation normal,*
*based on at least 10 years of data in GPCC data base (*Schneider et al., 2014)

Huffman, G.J.,Bolvin, D.T.,Nelkin, E.J.: Day 1 IMERG Final Run Release Notes; NASA Goddard Earth Sciences Data and Information Services Center: Greenbelt, MD, USA, 2015.

Schneider, U., Becker, A., Finger, P. et al. GPCC's new land surface precipitation climatology based on quality-controlled in situ data and its role in quantifying the global water cycle. Theor Appl Climatol (2014) 115: 15. doi:10.1007/s00704-013-0860-x

2.	P.3 L.5: Two questions related to details: could the observed rainfall have been interpolated to a 0.25x0.25 grid for better comparison with the TMPA data sets? 0.15x0.15 for the GPM-IMERG dataset? And in regions of mountainous terrain, oftentimes consideration of the altitude-rainfall gradient relationship is critical for spatially distributing rainfall onto a grid. Was this information included in the interpolation? Do the authors think this effect is important in this region?

Initially, observed rainfall was interpolated for 0.5°*0.5°, 0.25°*0.25° and 0.15°*0.15° grids (without significant changes in the rainfall analysis or hydrological model performance). Nonetheless, not to lose more detailed spatial information a 0.10°*0.10° grid over Andean regions (in this paper greater than 1500 masl) was selected for the interpolation process, since there are more rainfall stations close to each other over Andean regions than Amazon regions (in this paper, region lower than 1500 masl).

3.	P.4 L.9: MGB uses the rainfall aggregated to a daily time step. So, it seems that one of the main possible advantages of the GPM-IMERG (30 minute time step) datatset compared to the TMPA-V7 (3 hourly) for computing runoff generation by MGB is lost, especially since the convective nature of the rainfall is likely better resolved, in theory at least, using a 30 minute time step. Is there anything in MGB that can take advantage of the diurnal temporal distribution of the rainfall? If not, I think the authors should at least comment that hydrological models using sub-diurnal time steps might have larger differences between the TMPA and GPM-IMERG datasets owing to their different temporal resolutions...Can the authors comment on this?

Thank you for your comment

The improvements in MGB-IPH software do not currently include topics about sub-daily time step (https://www.ufrgs.br/hge/mgb-iph/).

In general, the performance of the model when using GPM-IMERG datasets indicates that these data are useful for estimating observed streamflows in Andean-Amazonian regions (Ucayali basin, southern regions of the Peruvian and Ecuadorian Amazon Basin). These results are similar to those obtained from TMPA V7 estimates by Zubieta et al. [2015] for the 2003-2009 period. Streamflows obtained from GPM-IMERG, TMPA V7, TMPA RT datasets show the same spatial pattern as those obtained by using PLU, (low and high performance in the northern and southern regions of the ABPE, respectively). The ability to represent seasonal streamflows in the southern region using these four precipitation datasets is validated with statistical evaluation.

It is important to note that advantages of GPM-IMERG compared to the TMPA-V7, such as the temporal resolution (30 minutes against 3 hours, respectively), for estimating streamflows have not yet been fully analyzed. The use of sub-daily rainfall data is potentially interesting to simulate discharge variability in the Andean rivers, where short convective rainfall episodes are more relevant for hydrological variability. In this study, precipitation and streamflows were analyzed at the daily time step. Further flash flood modeling at smaller scales would be able to evidence the effects of sub-diurnal differences between datasets

Zubieta, R.,Geritana, A., Espinoza, J.C. and  Lavado W.: Impacts of Satellite-based Precipitation Datasets on Rainfall-Runoff Modeling of the Western Amazon Basin of Peru and Ecuador, Journal of Hydrology, doi:10.1016/j.jhydrol.2015.06.064, 2015.

4.	P.5 L.1: How were these thresholds selected? Are they based on some sort of statistical analysis?

The Amazon basin of Perú and Ecuador can present different rainfall regimes (Espinoza et al., 2009; Laraque et al., 2007). Rainfall thresholds for the initiation of events such as landslides or floods can be variable in

space and time, for example, extreme rainfall (amount) in one Andean region may be normal in Amazon region. In this study, those thresholds are obtained from frequency analysis (percentiles 5, 20, 60, 90, 95).

Espinoza, J.C, Ronchail, J., Guyot, J.L., Cochonneau, G., Filizola, N.P., Lavado, C., De Oliveira, E., Pombosa, R., Vauchel P.: Spatio-Temporal rainfall variability in the Amazon basin countries (Brazil, Peru, Bolivia, Colombia and Ecuador), I.J. of Climatology 29(11): 1574–1594, doi:10.1002/joc.1791, 2009.

Laraque, A., Ronchail, J., Cochonneau, G., Pombosa, R., Guyot, J.L.: Heterogeneous distribution of rainfall and discharge regimes in the Ecuadorian Amazon basin, Journal of Hydrometeorology 8: 1364–1381, doi:10.1175/2007JHM784.1, 2007.

5.      P.5 L.25: The text "overestimate observations"...should likely be modified to something like "produce overestimates compared to observations".

Thank you so much, that would be considered:

Total annual rainfall over the ABPE during the selected period is shown in Figs. 1c-f, using all four precipitation products. The satellite-based datasets (GPM-IMERG, TMPA V7 and TMPA RT) produce overestimates compared to observations (PLU) during this period (by 11.1%, 15.7% and 27.7 %, respectively).

6.      P.6 L.10-11: The authors have reported that the satellite-based datasets underestimate the dry and wet season rainfall much more for the Huallaga basin compared to the Ucayali basin: do the authors have any insights as to why this is?

The Huallaga basin is not predominantly an Amazon region as is the Ucayali basin. The Andean region (higher than 1500 msal) is more present in the Huallaga basin (51%) than Ucayali basin (39%). The limitations to represent adequate rainfall from the satellite-based precipitation can be due to the strong spatial variability of rainfall in the Amazon-Andes region. Our finding about predominant underestimation in relation to observed rainfall along the Huallaga basin is consistent with others research developed over Andean regions of Peru (Condom et al, 2010), Bolivia (Scheel et al., 2011) and Ecuador ( Zulkafli et al., 2014).

Condom, T., Rau, P., Espinoza, J.C. 2011. Correction of the TRMM 3B43 monthly precipitation data over the mountainous areas of Peru during the period 1998–2007. Hydrological Processes.DOI: 10.1002/hyp.7949.

Scheel, M. L. M., Rohrer, M., Huggel, Ch., Santos Villar, D., Silvestre, E., and Huffman, G. J. 2011. Evaluation of TRMM Multi-satellite Precipitation Analysis (TMPA) performance in the Central Andes region and its dependency on spatial and temporal resolution, Hydrol. Earth Syst. Sci., 15, 2649-2663, doi:10.5194/hess-15-2649.

Zulkafli, Z., Buytaert, W., Onof, C., Manf, B., Tarnavsky, E., Lavado, W., and Guyot, J. L. 2014. A Comparative Performance Analysis of TRMM 3B42 (TMPA) Versions 6 and 7 for Hydrological Applications over Andean–Amazon River Basins. J. Hydrometeor., 15, 581–592, DOI: 10.1175/JHM-D-13-094.1

7.      P.6 L.18-20: While I am not surprised that detection of light events was difficult, why do the products have such difficulty predicting strong rainfall events? I might have (perhaps erroneously or naively I admit) assumed that such events might be better detected. Can the authors comment on this? For example, are the strongest events occurring in high altitude/mountainous regions which are more difficult to detect? Is the smoothing to daily averages related to this problem?

Thank you for the suggestion, these paragraphs would be considered in the manuscript

High or extreme precipitation events can be variable in space and time, rainfall amount for extreme events in one Andean mountain may be normal in Amazon region. This limitation to represent adequate rainfall from the satellite-based precipitation can be due to the strong spatial variability of rainfall in the Amazon-Andes

region. Indeed, the AB is distinguished by complex rainfall spatial distribution from the interactions between topography and large-scale humidity transport [Espinoza et al., 2015].

Assessment of rainfall estimates (GPM-IMERG, TMPA V7 and TMPA RT) with respect to PLU have been also perfomed using the Heidke Skill Score (HSS). HSS is a measure of skill in predictions, classified as below normal, near-normal and above-normal (Wilks, 1995). The assesment from HSS is based on the number of correctly predicted data where the category with the largest probability turns out to be correct. As reflected in the formula: $HSS = \frac{C-E}{N-E}$ , where C is the number of correct predictions, E is the number of correct predictions expected by chance and N is the total number of predictions. HSS = 1 refers to a perfect prediction, HSS = 0 shows no skill and HSS < 0,  indicates that a prediction is worse than a random prediction.

The HSS spatial distribution estimated from daily precipitation using each satellite dataset (GPM-IMERG, TMPA V7 and TMPA RT) and PLU was calculated using thresholds (0.1, 1, 5, 10 and 20 mm/day) as a reference prediction (Fig. S3a-c). In general, for the daily scale, the HSS score varies between 0 and 0.4, indicating low skill. The mean HSS for GPM-IMERG shows a moderate HSS score of around 0.4 in the Northern region (Fig. S3a). The lowest HSS values (lower than 0.2) for GPM-IMERG are mainly located in the Andean regions, where there are more rainfall stations than in the Amazonian regions. This could be due to strong spatial variability, which is characterized by rainfall decrease with altitude and by the leeward or windward position of the stations (Espinoza et al, 2009). Low scores are also observed in more scattered areas along the ABPE when TMPA V7 and TMPA RT are analyzed (lower than 0.15). Nevertheless, this relationship is slightly improved in the northern region of the Ucayali basin (~0.2).

[Figure]

Fig. S3. Spatial variability of the Heidke Skill Score from a) GPM-IMERG, b) TMPA V7 and c) TMPA RT against PLU ground observation, period from 2014 to 2015.

Espinoza, J. C., Chavez, S, Ronchail, J., Junquas, C., Takahashi, K. and Lavado, W., 2015. Rainfall hotspots over the southern tropical Andes: Spatial distribution, rainfall intensity,and relations with large-scale atmospheric circulation. Water Resour.Res. 51, 3456-3475.

8.   P.7 L.3: The calibration is glossed over a bit it seems: what set of parameters were calibrated? Were parameters calibrated separately for each precipitation dataset?. Also, no information is given on the quality or calibration of the MGB evaporation.  Is it significant compared to the rainfall? Are there non-negligible compensating errors (evaporation bias might offset rainfall or discharge errors/biases)? A short discussion is needed.

To optimize the simulation of streamflows from precipitation datasets, different parameter sets were assigned to each basin in the ABPE during calibration. Analysis by sub-basin is more reliable than assigning the same parameter set to the entire basin [Zubieta et al., 2015]. Based on sensitivity analysis of the MGB-IPH model [Collischonn et al., 2007] six parameters were selected for calibration: $Wm_i$ (mm), $b_i$ (–), $Kint$ (mm.d$^{-1}$), $Kbas_i$ (mm.d$^{-1}$), $CS_i$ (–) and $CI_i$ (–), where Wm represents water retained in the soil, which influences the evaporation process over time; $Kint$ and $Kbas$ control the amount of water in cases in which subsurface soil and groundwater, respectively, are saturated; and $CS$ and $CI$ allow for adjustment of retention time of flows [Collischonn et al., 2007]. To determine optimal parameters, an automatic calibration process was used in order to reduce the domain extent; a previous manual adjustment of the values was performed. To ensure impartiality, parameter sets were calibrated separately for each precipitation dataset. for each parameter value, different domains were considered initially, in which a first value determined by manual calibration was defined as the relative centroid for each domain. The MOCOM-UA multi-criteria global optimization algorithm [Yapo et al., 1998] was then used to find optimal solutions for six parameters. This process results in an effective and efficient search on the Pareto optimum space [Boyle et al., 2000]. To analyze the impacts on the calibrated parameters, average parameters were calculated for precipitation datasets and HRU (Table 4).

The results of the calibration process indicate that overestimation by TMPA RT compared to observed rainfall (PLU), GPM-IMERG and TMPA V7 (Fig. 2a) in several months is consistent with a mean increase in $Wm$ (+53%, +6%, +15% respectively), along with a predominantly mean decrease in $Kbas$ (-18%, -39% and -16% respectively) and $Kint$ (-25%, -15%, +2%) to achieve water balance (Table 4). Meanwhile, the overestimation by PLU compared to GPM-IMERG, TMPA V7 and TMPA RT (Fig. 3a) is consistent with a mean increase in $Wm$ (+33%, +38%, +34% respectively), along with a mean decrease in $Kbas$ (-30%, -28% and -38% respectively) and $Kint$ (-17%, -16%, -17%) to achieve water balance (Table 4).

Table 3. Model parameters subjected to the process of automatic calibration for the Peruvian and Ecuadorian Amazon basin.

| Parameter | HRU | Hydrological process | First guess | Domain |
|---|---|---|---|---|
| Wm(mm) | Shrubs, agricultural areas/not deep soils | Water storage on the HRU | 200 | 50-1200 |
|  | Shrubs, agricultural areas/deep soils |  | 400 | 50-1200 |
|  | Forest/not deep soils |  | 350 | 50-1200 |
|  | Forest/deep soils |  | 600 | 50-1200 |
|  | Pasture/not deep soils |  | 120 | 50-1200 |
|  | Pasture/deep soils |  | 240 | 50-1200 |
| Kint(mm/d) | Shrubs, agricultural areas/not deep soils | Sub - surface flow | 80 | 50-150 |
|  | Shrubs, agricultural areas/deep soils |  | 90 | 50-150 |
|  | Forest/not deep soils |  | 100 | 50-150 |
|  | Forest/deep soils |  | 120 | 50-150 |
|  | Pasture/not deep soils |  | 70 | 50-150 |
|  | Pasture/deep soils |  | 80 | 50-150 |
| Kbas(mm/d) | Shrubs, agricultural areas/not deep soils | Groundwater flow | 30 | 10 - 100 |
|  | Shrubs, agricultural areas/deep soils |  | 50 | 10 - 100 |
|  | Forest/not deep soils |  | 70 | 10 - 100 |
|  | Forest/deep soils |  | 80 | 10 - 100 |
|  | Pasture/not deep soils |  | 55 | 10 - 100 |
|  | Pasture/deep soils |  | 70 | 10 - 100 |
| CS | All | Surface flow | 15 | 0.35 - 40 |
| CI(-) | All | Sub-surface flow | 120 | 1 - 200 |

| b(-) | All | Variable infiltration curve | 0.12 | 0.01 - 2 |
| --- | --- | --- | --- | --- |

Table 4. Values of the model mean parameters used in the Ucayali and Huallaga basins for each rainfall datasets for the 2014-2015 period.

| Parameter | HRU | UCAYALI BASIN | | | | HUALLAGA BASIN | | | |
| --- | --- | --- | --- | --- | --- | --- | --- | --- | --- |
| | | PLU | GPM-IMERG | TMPA V7 | TMPA RT | PLU | GPM-IMERG | TMPA V7 | TMPA RT |
| Wm(mm) | Shrubs, agricultural areas/not deep soils | 268 | 351 | 294 | 373 | 100 | 60 | 65 | 60 |
| | Shrubs, agricultural areas/deep soils | 340 | 472 | 503 | 597 | 132 | 102 | 96 | 99 |
| | Forest/not deep soils | 300 | 408 | 273 | 344 | 130 | 101 | 99 | 96 |
| | Forest/deep soils | 422 | 453 | 445 | 435 | 250 | 203 | 180 | 209 |
| | Pasture/not deep soils | 144 | 350 | 261 | 321 | 101 | 60 | 66 | 59 |
| | Pasture/deep soils | 196 | 400 | 454 | 496 | 150 | 120 | 116 | 121 |
| Kint (mm/d) | Shrubs, agricultural areas/not deep soils | 141 | 216 | 151 | 151 | 190 | 161 | 163 | 152 |
| | Shrubs, agricultural areas/deep soils | 180 | 236 | 156 | 163 | 220 | 189 | 195 | 198 |
| | Forest/not deep soils | 198 | 123 | 107 | 108 | 103 | 162 | 155 | 160 |
| | Forest/deep soils | 200 | 134 | 108 | 113 | 120 | 208 | 199 | 220 |
| | Pasture/not deep soils | 150 | 110 | 119 | 122 | 121 | 160 | 151 | 150 |
| | Pasture/deep soils | 180 | 113 | 126 | 128 | 132 | 193 | 201 | 190 |
| Kbas (mm/d) | Shrubs, agricultural areas/not deep soils | 103 | 121 | 89 | 93 | 55 | 70 | 72 | 80 |
| | Shrubs, agricultural areas/deep soils | 113 | 123 | 100 | 103 | 61 | 90 | 94 | 100 |
| | Forest/not deep soils | 53 | 134 | 59 | 53 | 44 | 70 | 69 | 80 |
| | Forest/deep soils | 62 | 25 | 69 | 62 | 63 | 90 | 88 | 100 |
| | Pasture/not deep soils | 64 | 112 | 66 | 64 | 46 | 70 | 76 | 80 |
| | Pasture/deep soils | 74 | 113 | 71 | 71 | 63 | 90 | 66 | 100 |
| CS | All | 18 | 16 | 17 | 17 | 2.6 | 2.4 | 2.6 | 2.5 |
| CI(-) | All | 112 | 111 | 118 | 111 | 111 | 133 | 135 | 132 |
| b(-) | All | 0.13 | 0.17 | 0.15 | 0.12 | 0.12 | 0.15 | 0.14 | 0.14 |

Yapo, P.O., Gupta, H.V., Sorooshian, S.:Multi-objective global optimization for hydrologic models. Journal of Hydrology 204, 83–97, 1998.

Boyle, D.P., Gupta, H.V., Sorooshian, S.: Toward improved calibration of hydrologic models: combining the strengths of manual and automatic methods. Water Resources Research 36 (12), 3663–3674, 2000.

Collischonn, W., Allasia, D.G., Silva, B.C., Tucci, C.E.M. : The MGB-IPH model for large-scale rainfall-runoff modeling, J. Hydrol. Sci. 52, 878–895, doi: 10.1623/hysj.52.5.878, 2007.

9.        P.9 L.8-9: The authors state that seasonal streamflows in the southern region are well modeled using the satellite datasets, and indeed the results support this conclusion. But in the northern part of the Western Amazon basin/region, the results seem to indicate that satellite products are not useful for obtaining streamflows from hydrological modeling: so this implies that further progress is still required.. It think this should also be stated in the conclusions.

Thank you so much, this paragraph would be included

In general, the performance of the model when using the GPM-IMERG dataset indicates that these data are useful for estimating observed streamflows in Andean-Amazonian regions (Ucayali basin, southern regions of the Peruvian and Ecuadorian Amazon Basin). These results are similar to those obtained from TMPA V7 estimates by Zubieta et al. [2015] for the 2003-2009 period. Streamflows obtained from the GPM-IMERG, TMPA V7 and TMPA RT datasets show the same spatial pattern as those obtained by using PLU (low and high performance in the northern and southern regions of the ABPE, respectively). The ability to represent seasonal streamflows in the southern region using these four precipitation datasets is validated with statistical evaluation. Low performance of the model identified in the northern region is mainly related to the lack of adequate rainfall estimates, because it is consistent with estimated streamflows, so this implies that further progress is still required in satellite estimates of rainfall.

Zubieta, R.,Geritana, A., Espinoza, J.C. and  Lavado W.: Impacts of Satellite-based Precipitation Datasets on Rainfall-Runoff Modeling of the Western Amazon Basin of Peru and Ecuador, Journal of Hydrology, doi:10.1016/j.jhydrol.2015.06.064, 2015.

10.    P.9 L.2.: A 20% detection rate seems low. Can the authors put this into some sort
of context for the reader (e.g. is 20% indeed a reasonable value in this region, or would one hope to have 50%? or a higher value?). Also, what are the implications of these statistics and their impact on the MGB model simulations? Can a lower, say 10% detection rate, be assumed to be able to produce reasonable Nash scores? Or perhaps there is no clear relationship between these scores and the modeled discharge quality? This is not clear. Can the authors comment on this?

Thank you, a paragraph would be included in manuscript

Analysis of rain events from pixel value comparing PLU and estimated daily rainfall (GPM-IMERG, TMPA V7 and TMPA RT) suggests a low capacity for detection. This does not imply that they are not useful for hydrological modeling, because rain events not correctly detected for a region or a day could be correctly detected on another day or in nearby regions, compensating for the estimation of rainfall amount over large regions.

11.        P10 L.15-19: In Fig.4d, it is seen that the poorest Nash scores are in the northern part of the region shown: looking at Fig.4a (or Fig.1b), we see that there are relatively few observations in this region. But, this is where one would hope to benefit the most from a satellite product, but it seems this is not the case. In L.15-19 it is stated that such products hold promise for operational applications in data sparse regions, but it doesn't seem to be the case here? Can the authors comment a bit more or perhaps modify this slightly (seemingly) over-optimistic text?

I am sorry, you are right. The comment would be modified

Their usefulness in Andean-Amazon basins and their applicability as input to hydrological models have been evaluated recently by comparing modeled and observed datasets. Results indicate that these datasets could be used for operational applications in some Andean-Amazon *regions* [Zulkafli et al., 2014; Zubieta et al., 2015]. However, hydrological modeling using satellite-based precipitation data does not yield successful results in equatorial regions. This is mainly because of inadequate satellite estimates, because streamflows resulting from hydrological modeling using observed rainfall show acceptable performance in the Napo River basin in the equatorial region [Zubieta et al., 2015].

Zubieta, R.,Geritana, A., Espinoza, J.C. and  Lavado W.: Impacts of Satellite-based Precipitation Datasets on Rainfall-Runoff Modeling of the Western Amazon Basin of Peru and Ecuador, Journal of Hydrology, doi:10.1016/j.jhydrol.2015.06.064, 2015.

Zulkafli, Z., Buytaert, W., Onof, C., Manf, B., Tarnavsky, E., Lavado, W., and Guyot, J. L.: A Comparative Performance Analysis of TRMM 3B42 (TMPA) Versions 6 and 7 for Hydrological Applications over Andean–Amazon River Basins, J.Hydrometeor., 15, 581–592, doi: 10.1175/JHM-D-13-094.1, 2014.

---

## Author Comment (AC2) · 15 Feb 2017

**We would like to thank the referee for his useful comments. Please find bellow our answers.**

Scarcity of the rain gauge network and impact on the comparison results The authors had to work with a very scarce and unevenly spread network – (scarcity which makes satellite information all the more attractive) . They acknowledge briefly that the small number of gauges in part of the basin might explain the discrepancies between the sat/ground rainfall products and between the simulated/observed discharge , but there is no attempt to quantify the uncertainty in the ground rainfall product.

1. the authors should elaborate on the abilty of their ground rainfall product PLU to reproduce the rainfall gradients in the mountaneous part of the basins. Is altitude taken into account in their interpolation method and if yes how and was the quantitative uncertainty assessed ? - Also as kriging provides the estimation variance, the authors could provide a map showing the expected quality of the ground product (for instance ratio of kriging std over rainfall estimate for one day or an average over the season)

[Figure]

Figure S1.a) Relationship between altitude (m asl) and the observed and interpolated (kriging-PLU) annual rainfall (mm) for the 181 stations of the Peruvian and Ecuadorian Amazon basin for the 2014-2015 period.

To evaluate the ability of PLU to reproduce rainfall gradients in the Andes, the relationship between annual rainfall and altitude for 181 stations was compared. In this area, 100 rainfall station are located above 2000 m asl; some record in excess of 1500 mm/year, while less than 1200 mm/year is generally recorded above 3000 m asl. At lower elevations, abundant rainfall is associated with warm, moist air and the release of a large quantity of water vapor over the first eastern slope of the Andes; as a result, the amount of rainfall decreases with altitude (Laraque et al., 2007; Espinoza et al., 2009). A group of 15 observed rainfall stations located above 2000 m asl shows rainfall amount below 450 mm/year; this group cannot be adequately represented by PLU. Despite these differences, PLU and observed average rainfall show similar behavior at similar altitudes (Fig. S1). Indeed, the observed average rainfall for 181 stations shows high correlation with PLU for the 2014-2015 period ($r = 0.77$ p<0.01) (Fig. S2a). In contrast, observed average rainfall shows lower correlation with GPM-IMERG, TMPA V7 and TMPA RT (0.6, 0.56 and 0.61, respectively) (Fig. S2b-d).

[Figure]

 Fig.S2. Regression line between the observed annual rainfall in 181 rainfall stations (OR) and annual rainfall obtained from a) interpolation (PLU), b) GPM-IMERG, c) TMPA V7, d) TMPA RT for the 2014-2015 period.

Espinoza, J.C, Ronchail, J., Guyot, J.L., Cochonneau, G., Filizola, N.P., Lavado, C., De Oliveira, E., Pombosa, R., Vauchel P.: Spatio-Temporal rainfall variability in the Amazon basin countries (Brazil, Peru, Bolivia, Colombia and Ecuador), I.J. of Climatology 29(11): 1574–1594, doi:10.1002/joc.1791, 2009.

Laraque, A., Ronchail, J., Cochonneau, G., Pombosa, R., Guyot, J.L.: Heterogeneous distribution of rainfall and discharge regimes in the Ecuadorian Amazon basin, Journal of Hydrometeorology 8: 1364–1381, doi:10.1175/2007JHM784.1, 2007.

2. - The authors have used krigging to provide a product at the 0.1_ resolution over the 700000 km2 basin, from a total of 181 gauges. It would be informative to know what the de-correlation distance of the variogram model is ?. -Is anisotropy considered when interpolating in the montaneous areas ?

The Andean region is considered during the interpolation process. Indeed, the maximum distance of the semi-variogram selected consider both Andean and Amazonian regions (11.8° - ~1300 km). When needed, data transformations and anisotropy considerations were applied. It is important to mention, there is more uncertainty (over northern Amazonian regions and southern Andean regions) when the de-correlation distance is higher than 800 km (~0.74°) in the interpolation process (example: a and b semi-variograms, respectively).

[Figure]

3. - The authors provide comparison of satellite/ground product for basin average ; they should indicate what are the results when comparing only over the grid points that contain a gauge (or are within a short distance from gauges).

Thank you for your comment, first (for entire the basin), a Heidke skill score map between PLU against satellite-based precipitation were evaluated

Comparison of rainfall estimates (GPM-IMERG, TMPA RT) to PLU has been also perfomed using the Heidke Skill Score (HSS). HSS is based on the number of correctly predicted data where the category with the largest probability proves to be correct, as reflected in the formula: $= \frac{C-E}{N-E}$ , where C is the number of correct predictions, E is the number of correct predictions expected by chance and N is the total number of predictions. HSS = 1 refers to a perfect prediction, HSS = 0 shows no skill and HSS < 0, indicates that a prediction is worse than a random prediction.

The HSS spatial distribution estimated from daily precipitation using each satellite dataset (GPM-IMERG, TMPA V7 and TMPA RT) and PLU was calculated using thresholds (0.1, 1, 5, 10 and 20 mm/day) as a reference prediction (Fig. S3a-c). In general, for the daily scale, the HSS score varies between 0 and 0.4, indicating low skill. The mean HSS for GPM-IMERG shows a moderate HSS score of around 0.4 in the Northern region (Fig. S3a). The lowest HSS values (lower than 0.2) for GPM-IMERG are mainly located in the Andean regions, where there are more rainfall stations than in the Amazonian regions. This could be due to strong spatial variability, which is characterized by rainfall decrease with altitude and by the leeward or windward position of the stations (Espinoza et al, 2009). Low scores are also observed in more scattered areas along the ABPE when TMPA V7 and TMPA RT are analyzed (lower than 0.15). Nevertheless, this relationship is slightly improved in the northern region of the Ucayali basin (~0.2).

[Figure]

Fig. S3. Spatial variability of the Heidke Skill Score from a) GPM-IMERG, b) TMPA V7 and c) TMPA RT against PLU ground observation, period from 2014 to 2015.

Second, despite these differences, PLU and observed average rainfall show similar behavior at similar altitudes (Fig. S1). Indeed, the observed average rainfall for 181 stations shows high correlation with PLU for the 2014-2015 period ($r = 0.77$ p<0.01) (Fig. S2a). In contrast, observed average rainfall shows lower correlation with GPM-IMERG, TMPA V7 and TMPA RT (0.6, 0.56 and 0.61, respectively) (Fig. S2b-d).

[Figure]

Fig.S2. Regression line between the observed annual rainfall in 181 rainfall stations (OR) and annual rainfall obtained from a) interpolation (PLU), b) GPM-IMERG, c) TMPA V7, d) TMPA RT for the 2014-2015 period.

4. - Given the points above, how certain are you that satellite products are overestimating rainfall (p 5 -section 4.1) rather than the ground based product underestimating it.

We assume that PLU is the interpolated key information (kriging) from rain gauge, which is compared to other satellites (GPM-IMERG, TMPA V7 and TMPA RT). The error of the kriging interpolation can be represented for a rainfall station as the value of the point minus the predicted value, i.e. the value on the linear regression, each point on the sub-panel Fig. S2a corresponds to a rainfall station. It is possible observing deficiencies in the KRIGING estimation, this is the case in regions with annual precipitation less than 1650 mm / year (box "a", predominant underestimation) and regions with precipitation greater than 1650 mm / year (box "b", predominant overestimation). However, rainfall obtained from Kriging interpolation method provides a better similitude with observed rainfall. (r = 0.77 p<0.01) (Fig. S2a) than other products.

[Figure]

Fig.S2a. Regression line between the observed annual rainfall in 181 rainfall stations (OR) and annual rainfall obtained from a) interpolation (PLU)

5. – information on model calibration and sources of uncertainty in the model run.
Section 4.2 :

- A description of the model configuration is lacking – the size of the HRU – and a
discussion on whether or not it allows to take advantage of the products improved
spatial resolution is missing.

A HRU (hydrological response unit) [Kouwen et al., 1993) approach is used to perform soil water balance by mean spatial classification of all areas with a similar combination of soil and land cover. The benefit of using HRUs is the increased accuracy in streamflow simulations at smaller scales, as they make it possible to take better advantage of high spatial resolution databases for hydrological modeling applications. To create HRUs, the watershed is divided into regular elements (cells), which are interconnected by channels. A parameter set is calculated separately for each HRU of each pixel, considering only one layer of soil [Collischonn et al., 2007]. To reduce computational time, HRUs for small areas of the ABPE surface have been merged into those composing more representative areas. Finally, The ABPE was discretized for six HRUs into 2709 by 4533 pixels (400 m spatial resolution), it allows to take advantage of the products improved like GPM-IMERG (0.1° - ~11 km spatial resolution)

Kouwen, N. & Mousavi, S. F.: WATFLOOD/SPL9: Hydrological model and flood forecasting system. In: Mathematical Models of Large Watershed Hydrology (ed. by V. P. Singh & D. K. Frevert), Water Resources Publications. Highlands Ranch. Colorado, USA, 2002.

Collischonn, W., Allasia, D.G., Silva, B.C., Tucci, C.E.M. : The MGB-IPH model for large-scale rainfall-runoff modeling, J. Hydrol. Sci. 52, 878–895, doi: 10.1623/hysj.52.5.878, 2007.

6.- How was the model calibrated and on which period/data sets ? is the model recalibrated for each rainfall forcing ? if not/yes, why ?

Ok these paragraph and Tables would be considered

To optimize the simulation of streamflows from precipitation datasets, different parameter sets were assigned to each basin in the ABPE during calibration. Analysis by sub-basin is more reliable than assigning the same parameter set to the entire basin [Zubieta et al., 2015]. Based on sensitivity analysis of the MGB-IPH model

[Collischonn et al., 2007], six parameters were selected for calibration: $Wm_i$ (mm), $b_i$ (–), $Kint$ ($mm.d^{-1}$), $Kbas_i$ ($mm.d^{-1}$), $CS_i$ (–) and $CI_i$ (–), where Wm represents water retained in the soil, which influences the evaporation process over time; $Kint$ and $Kbas$ control the amount of water in cases in which subsurface soil and groundwater, respectively, are saturated; and $CS$ and $CI$ allow for adjustment of retention time of flows [Collischonn et al., 2007]. To determine optimal parameters, an automatic calibration process was used in order to reduce the domain extent; a previous manual adjustment of the values was performed. To ensure impartiality, parameter sets were calibrated separately for each precipitation dataset. Different domains were considered initially for each parameter value, and a first value, determined by manual calibration, was defined as the relative centroid for each domain. The MOCOM-UA multi-criteria global optimization algorithm [Yapo et al., 1998] was then used to find optimal solutions for six parameters. This process results in an effective and efficient search on the Pareto optimum space [Boyle et al., 2000]. To analyze the impacts on the calibrated parameters, average parameters were calculated for precipitation datasets and HRU (Table 4).

The results of the calibration process indicate that overestimation by TMPA RT compared to observed rainfall (PLU), GPM-IMERG and TMPA V7 (Fig. 2a) in several months is consistent with a mean increase in $Wm$ (+53%, +6%, +15% respectively), along with a predominantly mean decrease in $Kbas$ (-18%, -39% and -16% respectively) and $Kint$ (-25%, -15%, +2%) to achieve water balance (Table 4). Meanwhile, the overestimation by PLU compared to GPM-IMERG, TMPA V7 and TMPA RT (Fig. 3a) is consistent with a mean increase in $Wm$ (+33%, +38%, +34% respectively), along with a mean decrease in $Kbas$ (-30%, -28% and -38% respectively) and $Kint$ (-17%, -16%, -17%) to achieve water balance (Table 4).

Table 3. Model parameters subjected to the process of automatic calibration for the Peruvian and Ecuadorian Amazon basin.

| Parameter | HRU | Hydrological process | First guess | Domain |
|---|---|---|---|---|
| Wm(mm) | Shrubs, agricultural areas/not deep soils | Water storage on the HRU | 200 | 50-1200 |
| | Shrubs, agricultural areas/deep soils | | 400 | 50-1200 |
| | Forest/not deep soils | | 350 | 50-1200 |
| | Forest/deep soils | | 600 | 50-1200 |
| | Pasture/not deep soils | | 120 | 50-1200 |
| | Pasture/deep soils | | 240 | 50-1200 |
| Kint(mm/d) | Shrubs, agricultural areas/not deep soils | Sub - surface flow | 80 | 50-150 |
| | Shrubs, agricultural areas/deep soils | | 90 | 50-150 |
| | Forest/not deep soils | | 100 | 50-150 |
| | Forest/deep soils | | 120 | 50-150 |
| | Pasture/not deep soils | | 70 | 50-150 |
| | Pasture/deep soils | | 80 | 50-150 |
| Kbas(mm/d) | Shrubs, agricultural areas/not deep soils | Groundwater flow | 30 | 10 - 100 |
| | Shrubs, agricultural areas/deep soils | | 50 | 10 - 100 |
| | Forest/not deep soils | | 70 | 10 - 100 |
| | Forest/deep soils | | 80 | 10 - 100 |
| | Pasture/not deep soils | | 55 | 10 - 100 |
| | Pasture/deep soils | | 70 | 10 - 100 |
| CS | All | Surface flow | 15 | 0.35 - 40 |
| CI(-) | All | Sub-surface flow | 120 | 1 - 200 |
| b(-) | All | Variable infiltration curve | 0.12 | 0.01 - 2 |

Table 4. Values of the model mean parameters used in the Ucayali and Huallaga basins for each rainfall datasets  for the 2014-2015 period.

| Parameter | HRU | UCAYALI BASIN | | | | HUALLAGA BASIN | | | |
|---|---|---|---|---|---|---|---|---|---|
| | | PLU | GPM-IMERG | TMPA V7 | TMPA RT | PLU | GPM-IMERG | TMPA V7 | TMPA RT |
| Wm(mm) | Shrubs, agricultural areas/not deep soils | 268 | 351 | 294 | 373 | 100 | 60 | 65 | 60 |
| | Shrubs, agricultural areas/deep soils | 340 | 472 | 503 | 597 | 132 | 102 | 96 | 99 |
| | Forest/not deep soils | 300 | 408 | 273 | 344 | 130 | 101 | 99 | 96 |
| | Forest/deep soils | 422 | 453 | 445 | 435 | 250 | 203 | 180 | 209 |
| | Pasture/not deep soils | 144 | 350 | 261 | 321 | 101 | 60 | 66 | 59 |
| | Pasture/deep soils | 196 | 400 | 454 | 496 | 150 | 120 | 116 | 121 |
| Kint | Shrubs, agricultural areas/not deep soils | 141 | 216 | 151 | 151 | 190 | 161 | 163 | 152 |
| (mm/d) | Shrubs, agricultural areas/deep soils | 180 | 236 | 156 | 163 | 220 | 189 | 195 | 198 |
| | Forest/not deep soils | 198 | 123 | 107 | 108 | 103 | 162 | 155 | 160 |
| | Forest/deep soils | 200 | 134 | 108 | 113 | 120 | 208 | 199 | 220 |
| | Pasture/not deep soils | 150 | 110 | 119 | 122 | 121 | 160 | 151 | 150 |
| | Pasture/deep soils | 180 | 113 | 126 | 128 | 132 | 193 | 201 | 190 |
| Kbas | Shrubs, agricultural areas/not deep soils | 103 | 121 | 89 | 93 | 55 | 70 | 72 | 80 |
| (mm/d) | Shrubs, agricultural areas/deep soils | 113 | 123 | 100 | 103 | 61 | 90 | 94 | 100 |
| | Forest/not deep soils | 53 | 134 | 59 | 53 | 44 | 70 | 69 | 80 |
| | Forest/deep soils | 62 | 25 | 69 | 62 | 63 | 90 | 88 | 100 |
| | Pasture/not deep soils | 64 | 112 | 66 | 64 | 46 | 70 | 76 | 80 |
| | Pasture/deep soils | 74 | 113 | 71 | 71 | 63 | 90 | 66 | 100 |
| CS | All | 18 | 16 | 17 | 17 | 2.6 | 2.4 | 2.6 | 2.5 |
| CI(-) | All | 112 | 111 | 118 | 111 | 111 | 133 | 135 | 132 |
| b(-) | All | 0.13 | 0.17 | 0.15 | 0.12 | 0.12 | 0.15 | 0.14 | 0.14 |

Yapo, P.O., Gupta, H.V., Sorooshian, S.:Multi-objective global optimization for hydrologic models. Journal of Hydrology 204, 83–97, 1998.

Boyle, D.P., Gupta, H.V., Sorooshian, S.: Toward improved calibration of hydrologic models: combining the strengths of manual and automatic methods. Water Resources Research 36 (12), 3663–3674, 2000.

Collischonn, W., Allasia, D.G., Silva, B.C., Tucci, C.E.M. : The MGB-IPH model for large-scale rainfall-runoff modeling, J. Hydrol. Sci. 52, 878–895, doi: 10.1623/hysj.52.5.878, 2007.

Zubieta, R.,Geritana, A., Espinoza, J.C. and  Lavado W.: Impacts of Satellite-based Precipitation Datasets on Rainfall-Runoff Modeling of the Western Amazon Basin of Peru and Ecuador, Journal of Hydrology, doi:10.1016/j.jhydrol.2015.06.064, 2015.

7. - One of the benefit expected from new rainfall product like i-merg is their improved space/time resolution compared to coarser products. This important point is not discussed in the study. As the model is run a daily time step the benefit of improved time resolution cannot be assessed, however the authors could investigate the impact of the 0.1_ grid provided by i-merg. For instance by smoothing or under sampling the product to a coarser resolution (0_5 for instance). And comparing the simulated discharge for both 0_1 and a coarser spatial resolution.

To analyze the new benefits using GPM-IMERG in the hydrological modeling (*0.1*-degree by *0.1*-degree *spatial resolution*, while TMPA has 0.25°*0.25° *spatial resolution*) both small (< 20,000 Km$^2$) and large (> 20,000 km$^2$) basins were modeled. For example: Drainage areas (< 20,000 Km$^2$) controlled at Mejorada and KM105 stations using GPM-IMERG in the hydrological modeling are approximately evaluated from 134 and 77 pixels, respectively. Meanwhile, TMPA only approximately provides 24 and 16 pixels respectively.

Our results indicate that hydrological modeling are better using GPM- IMERG (NS=0.90) than TMPA V7 and TMPA RT (NS = 0.80 and 0.68, respectively) in drainage area controlled at KM105 station. However, results are more similar between them (NS ~0.65) in drainage are controlled at Mejorada station. It is important to note, that results of hydrological modeling using satellite-based precipitation datasets are better when small basins are assessed in Ucayali basin.

It is important to note that the advantages of GPM-IMERG over TMPA-V7 for estimating streamflows, such as temporal resolution (30 minutes compared to 3 hours, respectively), have not yet been fully analyzed. The use of sub-daily rainfall data can be potentially useful for simulating discharge in the Andean rivers, where short convective rainfall episodes are more relevant for hydrological variability. In this study, precipitation and streamflows were analyzed at a daily time step. Further flash flood modeling at smaller scales would reveal the effects of sub-diurnal differences between datasets.

8. - Rainfall is not the only source of uncertainty in the simulated discharge ; the ability of the chosen model to represent the hydrological processes in the studied region, especially in the mountainous sub-basins, should be discussed. Other sources of uncertainty–among them model parameters estimation- that might impact the results should also be mentioned and if they have been quantified, the information should be provided.

Thank you so much, this comment has been added in the manuscript

It is important to note that advantages of GPM-IMERG compared to the TMPA-V7, such as the temporal resolution (30 minutes against 3 hours, respectively), for estimating streamflows have not yet been fully analyzed. The use of sub-daily rainfall data is potentially interesting to simulate discharge variability in the Andean rivers, where short convective rainfall episodes are more relevant for hydrological variability. In this study, precipitation and streamflows were analyzed at the daily time step. Further flash flood modeling at smaller scales would be able to evidence the effects of sub-diurnal differences between datasets. Errors in streamflow simulations are mostly associated to input data uncertainty, including rainfall, limited representations of physical processes in models, and parameters such as DEM and HRUs. However, results show that it is possible to employ remote sensing data to large-scale hydrological models for streamflow simulations.

9. - importantly I could not find the information on the version of i-merg which has been

used here.

GPM (product final IMERG-V03D was considered),

10. - Since there were several releases of i-merg since the launch and given that i-merg (just like 3B42) is provided with a gauge calibrated version and an un-calibrated or RT version, both should be tested here. For a fair comparison with 3B42 RT and v7.

GPM-IMERG provides data :

• The Day-1 IMERG Final Run data sets (for the GPM era, mid-March 2014 to the present, delayed about 3 months) were released in late December 2014.

• The IMERG Late Run data sets begin 7 March 2015, while the Early Run start 1 April 2015.

We tried to expand the analysis period (more than June 2015), nonetheless, we had some disadvantages with data availability of this study (streamflow and rainfall gauges) to evaluate for example Early and Late Run products (to include calibration and validation period). We are sorry.

11. - P3 – line 15 to 35 – The degree of information on products should be the same for both i-merg and 3B42 : number /type of contribution satellites, basic description of the stimation method.

GPM is an international US/Japanese Earth science mission involving NASA and JAXA, respectively. The GPM mission improved and expanded on TRMM. GPM and TRMM provide precipitation data derived from different passive microwave (PMW) sources used in IMERG and TMPA, respectively [Huffman et al. 2015], including: Sounder for Atmospheric Profiling of Humidity in the Intertropics by Radiometry (SAPHIR), Advanced Technology Microwave Sounder (ATMS), Atmospheric Infrared Sounder (AIRS), Cross-Track Infrared Sounder (CRIS), and TRMM Combined Instrument (TCI) algorithms (2B31). They also include TRMM Microwave Image (TMI, data ended on 8 Apr 2015), GPM Microwave Imager (GMI), Advanced Microwave Scanning Radiometer for Earth Observing Systems (AMSR-E), Special Sensor Microwave Imager/Sounder (SSMIS), Microwave Humidity Sounder (MHS), Special Sensor Microwave Imager (SSM/I), Advanced Microwave Sounding Unit (AMSU), Operational Vertical Sounder (TOVS) and microwave-adjusted merged geo-infrared (IR). The precipitation datasets used in this study are as follows:

a)  GPM (product IMERG-V03D) data at several levels of processing have been provided since March 2014 (GPM-IMERG data are available at http://pmm.nasa.gov/GPM). The input precipitation estimates are computed using raw satellite measurements, such as those from passive microwave sensors (TMI, AMSR-E, SSM/I, SSMIS, AMSU, MHS, SAPHIR, GMI, ATMS, TOVS, CRIS and AIRS), inter-calibrated to the GPM Combined Instrument (GCI, using GMI and Dual-frequency Precipitation Radar, DPR) and adjusted with monthly surface precipitation gauge analysis data (where available). All these datasets are used to obtain the best estimate of global precipitation maps. The temporal resolution of IMERG-V03D is *half-hourly,* and it has a 0.1-degree by 0.1-degree spatial resolution. Unlike other satellites, such as TRMM, GPM-IMERG can detect both light and heavy rain and snowfall.

b)  TMPA 3B42 version 7 is obtained from the preprocessing of data provided by different satellite-based sensors between 1998 and April 2015, in both real and near-real time (TMPA 3b42 data are available at ftp://disc2.nascom.nasa.gov/data/TRMM/Gridded/3B42RT). The 3B42 algorithm (every three hours) combines precipitation estimates from TMI, AMSR, SSMIS, SSM/I, AMSU, MHS, TCI, *MetOp-B* and IR. After the preprocessing is complete, the 3-hourly multi-satellite estimations are summed for the month and combined with monthly rainfall obtained from Global Precipitation Climatology Centre (GPCC), which uses ground-based precipitation. The last step is to scale each 3-hourly rainfall estimate for the month to sum to the monthly value (for each pixel separately, 0.25-degree by 0.25-degree spatial resolution).

c) TMPA RT (real time) precipitation data are related to TMPA V7, but do not include calibration measurements of rainy seasons, which are incorporated more than a month after the satellite data. (ftp://disc2.nascom.nasa.gov/data/TRMM/Gridded/3B42RT). As with TMPA V7, the final, gridded, sub-daily temporal resolution of TMPA RT is usually every three hours, *with* a 0.25-degree by 0.25-degree spatial resolution.

d) To evaluate satellite-based datasets, a precipitation product was obtained using daily data series (PLU) from SENAMHI rainfall stations. We collected daily rainfall data for 202 rain stations during the selected period. Quality control based on the Regional Vector Method (RVM) was used to select stations having the lowest probability of errors in their data series [Hiez 1977; Brunet-Moret 1979]. Finally, 181 RVM-approved rainfall data series [distributed over 700,000 km$^2$] were selected, with data between March 2014 and June 2015 (Fig. 1b). The area with the highest data availability covers around 81% of the ABPE (19% without availability is mainly located in the northern region), where the largest distribution of rainfall stations is in the Andean regions, rather than Amazonian regions, of the Ucayali and Huallaga basins (the Huallaga is a sub-basin of the Marañón basin). For comparison, both regions with and without availability of rainfall data were considered for hydrological modeling. Rainfall observations subsequently were spatially interpolated to a resolution of $0.1° \times 0.1°$ by ordinary kriging, and a spherical semivariogram model was used to generate a gridded daily rainfall dataset. Data transformations and anisotropy were applied when necessary. This method has been used to interpolate environmental variables, such as rainfall in the Amazon and Andean regions (Guimberteau et al., 2012; Zubieta et al., 2016). To use each precipitation dataset as input to the hydrological model, sub-daily data (for example, TMPA datasets have temporal resolution of 3 hours) were rescaled to a daily time step.

Guimberteau, M., Drapeau, G., Ronchail, J., Sultan, B., Polcher, J., Martinez, J.M., Prigent, C., Guyot, J.L., Cochonneau, G., Espinoza, J.C., Filizola, N., Fraizy, P., Lavado, W., De Oliveira, E., Pombosa, R., Noriega, L., Vauchel, P.: Discharge simulation in the sub-basins of the Amazon using ORCHIDEE forced by new datasets. Hydrol. Earth Syst. Sci. 16, 911–935, 2012.

Huffman, G.J.,Bolvin, D.T.,Nelkin, E.J.: Day 1 IMERG Final Run Release Notes; NASA Goddard Earth Sciences Data and Information Services Center: Greenbelt, MD, USA, 2015.

Zubieta, R., Saavedra, M., Silva, Y., Giraldez, L.: Spatial analysis and temporal trends of daily precipitation concentration in the Mantaro River basin - Central Andes of Peru, Stochastic Environmental Research and Risk Assessment. DOI :10.1007/s00477-016-1235-5, 2016.

Brunet-Moret, Y.: Homogénéisation des précipitations. Cahiers ORSTOM, Série Hydrologie 16: 3–4, 1979.

Hiez, G. : L'homogénéité des données pluviométriques, Cahier ORSTOM, série Hydrologie 14: 129–172, 1977. Huffman, G., Adler, R., Bolvin, D., Gu, G., Nelkin, E., Bowman, K., Hong, Y., Stocker, E., Wolff, D.: The TRMM Multisatellite Precipitation Analysis (TCMA): quasi-global, multiyear, combined- sensor precipitation estimates at fine scales, Journal of Hydrometeorology 8, 38–55, doi:10.1.1.532.5634, 2007.

12. - 3B42 (and i-merg calibrated version) use some gauges from weather services – Could you check which are the gauges used here were included in TMPA/i-merg correction stage ?

Both TMPA and GPM-IMERG adopt the Global Precipitation Climatology Centre (GPCC) monthly rain gauge analysis (Huffman et al. 2015). The Monitoring Product is represented on internationally exchanged meteorological data i.e. gauge observations from world-wide 6,000 to 7,000 stations (see next figure, *(*Schneider et al., 2015). The average gauge density is about 2 gauges per 2.5° by 2.5° lat/long grid box only. Building upon the figure of rainfall stations and lat/long grid *(*Schneider et al., 2014), it very probably that 105 rainfall stations used in our study were considered by GPCC calculations.

[Figure]

*Spatial distribution of monthly in-situ stations with a climatological precipitation normal,*
*based on at least 10 years of data in GPCC data base (*Schneider et al., 2014).

Huffman, G.J.,Bolvin, D.T.,Nelkin, E.J.: Day 1 IMERG Final Run Release Notes; NASA Goddard Earth Sciences Data and Information Services Center: Greenbelt, MD, USA, 2015.

Schneider, U., Becker, A., Finger, P. et al. GPCC's new land surface precipitation climatology based on quality-controlled in situ data and its role in quantifying the global water cycle. Theor Appl Climatol (2014) 115: 15. doi:10.1007/s00704-013-0860-x

Other minor corrections :

13. -In the introduction and throughout the text there seem to be some confusion between
i) the satellite themselves (TRMM or GPM core satellite)

Thanks, this paragraph would be improved as well:

The aim of this paper is to evaluate the use of rainfall estimates from the GPM-IMERG in obtaining streamflows over the Amazon Basin of Peru and Ecuador (ABPE) during a 16-month period (from March 2014 to June 2015) when all datasets are available. It provides a comparative analysis of the GPM-IMERG, TMPA RT and TMPA V7 datasets with ground-based precipitation dataset (PLU). PLU was developed by spatial interpolation using the Peruvian National Meteorology and Hydrology Service (SENAMHI) network. Each precipitation dataset was used as input for the MGB-IPH hydrological model [Collischonn et al., 2007], which was recently adapted to ABPE [Zubieta et al., 2015].

Zubieta, R.,Geritana, A., Espinoza, J.C. and  Lavado W.: Impacts of Satellite-based Precipitation Datasets on Rainfall-Runoff Modeling of the Western Amazon Basin of Peru and Ecuador, Journal of Hydrology, doi:10.1016/j.jhydrol.2015.06.064, 2015.

Collischonn, W., Allasia, D.G., Silva, B.C., Tucci, C.E.M. : The MGB-IPH model for large-scale rainfall-runoff modeling, J. Hydrol. Sci. 52, 878–895, doi: 10.1623/hysj.52.5.878, 2007.

14. ii) satellite constellations (the GPM international program includes the NASA-JAXA GPM core satellite -
and TRMM while it was still going- and a constellation of other satellites from various agencies)

Thank you, this comment was considered

GPM is an international US/Japanese Earth science mission with the agencies of NASA and JAXA, respectively. GPM is an improved and expanded mission to TRMM. GPM and TRMM provide precipitation data derived from passive microwave (PMW) sources used in IMERG and TMPA, respectively [Huffman et al. 2015] …..

Huffman, G., Adler, R., Bolvin, D., Gu, G., Nelkin, E., Bowman, K., Hong, Y., Stocker, E., Wolff, D.: The TRMM Multisatellite Precipitation Analysis (TCMA): quasi-global, multiyear, combined- sensor precipitation estimates at fine scales, Journal of Hydrometeorology 8, 38–55, doi:10.1.1.532.5634, 2007.

15.  iii) the rainfall products which are derived from this satellite constellations and do not depend on one particular satellite (TMPA 3B42 can be run without the TRMM satellite itself: : : i-merg could be run without GPMcore if necessary ) . Exemple : intro p2 line 12-13 : 'satellite data sets : : :..are uniformly distributed in space and time' - Product like 3B42 or i-merg are provided on a regular space-time grid however the microwave satellite information itself is provided with a very irregular sampling and depends on individual orbits: : :. And as for gauges interpolation is done to prvide a final regularly gridded product.

I am sorry, you are right, the comment would be improved

Satellite-based datasets uniformly distributed in both space and time offer an alternative for modeling hydrological events

16. P 2 – line 30-32 – confusion between GPMcore single satellite launched in 2014, and the GPM multi satellite/multiagencies constellation : : :..

Also : The improved resolution capacity of the latest products does not come from a specific satellite (though some members of the GPM constellation such as TRMM and more recently Megha-Tropiques , provide additional sampling specifically in the Tropics) but from the overall sampling capacity of the whole constellation. This should be mentioned more clearly in the intro.

Thank you, this paragraph would be included

The GPM mission [Schwaller and Morris, 2011], launched in February 2014, comprises an international constellation of satellites that provide rainfall estimations with significant improvements in spatio-temporal resolution, compared to TMPA products. This is true of GPM products such as Integrated Multi-satellite Retrievals (IMERG) estimations. Recent studies highlight that the GPM-IMERG estimations can adequately substitute for TMPA estimations both hydrologically and statistically, despite limited data availability [Liu, 2016; Tang et al., 2016].

Schwaller, M. R. and K. R. Morris.: A Ground Validation Network for the Global Precipitation Measurement Mission, J. Atmos. Oceanic Technol., 28, 301–319, doi: 10.1175/2010jtecha1403.1. 2011.

Liu, Z.: Comparison of Integrated Multisatellite Retrievals for GPM (IMERG) and TRMM Multisatellite Precipitation Analysis (TMPA) monthly precipitation products: Initial results,  J. Hydrometeor., 17, 777–790, doi:10.1175/JHM-D-15-0068.1, 2016.

Tang, G., Z. Zeng, D. Long,X.Guo, B.Yong,W. Zhang, and Y. Hong.: Statistical and hydrological comparisons between TRMM and GPM level-3 products over a midlatitude basin: Is day-1 IMERG a good successor for TMPA 3B42V7?,  J. Hydrometeor., 17, 121–137, doi:10.1175/jhm-d-15-0059.1, 2016.